# Crash-perching on vertical poles with a hugging-wing robot
**Mohammad Askari** [1] ✉, **Michele Benciolini** [1], **Hoang-Vu Phan** [1], **William Stewart** [1,2], **Auke J. Ijspeert**[3] & **Dario Floreano** [1]

Perching with winged Unmanned Aerial Vehicles has often been solved by means of complex control or intricate appendages. Here, we present a method that relies on passive wing morphing for crash-landing on trees and other types of vertical poles. Inspired by the adaptability of animals' and bats' limbs in gripping and holding onto trees, we design dual-purpose wings that enable both aerial gliding and perching on poles. With an upturned nose design, the robot can passively reorient from horizontal flight to vertical upon a head-on crash with a pole, followed by hugging with its wings to perch. We characterize the performance of reorientation and perching in terms of impact speed and angle, pole material, and size. The robot robustly reorients at impact angles above 15° and speeds of $3 \, \text{m} \cdot \text{s}^{-1}$ to $9 \, \text{m} \cdot \text{s}^{-1}$, and can hold onto various pole types larger than 28% of its wingspan in diameter. We demonstrate crash-perching on tree trunks with an overall success rate of 73%. The method opens up new possibilities for the use of aerial robots in applications such as inspection, maintenance, and biodiversity conservation.

Winged Unmanned Aerial Vehicles (UAVs) are particularly suitable for long-distance missions, such as delivery, mapping, and search and rescue, as they offer higher endurance per mass compared to other types of UAVs[1]. However, compared to winged flying animals, they have limited ability to land or perch on complex structures for tasks like inspection, manipulation, monitoring, or battery recharging[2]. This limitation has spurred the development of control and mechanical systems to enable perching[3]. Inspired by avian perching[4], most control-oriented studies have predominantly focused on pitch-up maneuvers and post-stall control for reducing speed at landing[5–7], and used either microspines to perch the winged UAV on a vertical wall[2,8] or hooks to hang a glider on a cable[9,10]. However, executing such maneuvers requires sensory systems embedded with control algorithms to ensure high level of accuracy within a short time frame. Furthermore, the rapid pitch-up maneuver is prone to potentially dangerous flight conditions due to reduced control effectiveness and aerodynamic stall at low speeds with high angles of attack.

In order to avoid the complex pitch-up maneuver, solutions with mechanical systems have been proposed as alternatives. Mechanical solutions for perching with hover-capable multicopters abound. Examples include passive avian-inspired claws[11,12] and wrapping-arms[13,14] for perching on branches, microspines for holding onto flat, rough, vertical surfaces[15], modular landing gear[16], active[17] and passive[18–20] compliant grippers, dry adhesive and fiber-based pads[21–23], and spider-like perching using threaded anchors[24]. However, there are fewer options available for winged UAVs.

Anderson et al.[25] developed a simple adhesive-based mechanism that allows a fixed-wing UAV to attach to vertical surfaces upon head-on impact, followed by hanging from the anchor with a tether. Like most glue-based attachment concepts, this technique is surface-dependent and may not function effectively on damp or dusty surfaces. The study also does not provide any formal characterization of perching performance. Kovač et al.[26] proposed a system for perching a very lightweight microglider on walls, which consists of spring-loaded needles driven into the wall upon direct impact at speeds of up to $4 \, \text{m} \cdot \text{s}^{-1}$. However, this approach is more suitable for very lightweight robots with a weight of several tens of grams. For larger-scale systems, Stewart et al.[27] introduced a passive perching claw that can mitigate kinetic energy to enable crash-perching at speeds up to $7.4 \, \text{m} \cdot \text{s}^{-1}$. Although the proposed claw design worked well for perching and hanging on small horizontal bars up to 55 mm, it is not easily scalable or applicable to perching on vertical poles with larger diameters.

Here, we propose a method for the passive perching of winged UAVs on vertical poles, which are ubiquitous in man-made environments such as building scaffolding, electric towers, street lights, and utility poles. Perching on natural poles, such as trees, could also be helpful in biodiversity conservation or wildlife monitoring[13,28]. Geckos in their natural habitat exhibit a remarkable landing strategy on tree trunks. They crash head-first onto the trunk followed by a full body rotation, which is halted by the landing of their hind limbs and tail[29]. Inspired by the gecko's touchdown technique, our proposed method incorporates an "upturned nose" element that allows

[1]Laboratory of Intelligent Systems, EPFL, Lausanne CH-1015, Switzerland. [2]Soft Flyers Group, Stony Brook University, New York 11794 NY, USA. [3]Biorobotics Laboratory, EPFL, Lausanne CH-1015, Switzerland. ✉e-mail: mohammad.askari@epfl.ch

passive reorientation from horizontal flight to a vertical attitude upon impact with the pole, thus foregoing control of complex pitch-up maneuvers at near-stall angles of attack. The UAV then leverages foldable, pre-loaded segmented wings, which are released through a latch system at impact, to wrap around vertical poles for perching. This behavior imitates that observed in certain flying animals (see Fig. 1). While studies have examined morphological adaptations and use of wings, feet, and tails of select bat[30] and bird[31–33] species, a broader observation reveals common principles among all perching and climbing animals[34]. Our solution avoids dedicated perching feet structures that increase body mass and complexity, opting instead for a dual-use strategy leveraging existing UAV elements. This includes employing front limbs (wings) to tightly hug poles and maintain the center of mass close to the pole to minimize the pitch-back effect (Fig. 1). The use of a long tail is also found to be effective in nature for both landing[29] and resting[31,32]. Moving forward, we provide the design details of the upturned nose and wing elements, investigate the performance of inertial reorientation and wing wrapping induced by collisions, and validate the crash-perching capability on tree trunks using PercHug, a gliding-winged robot.

## Results

### Operating principle and robot design

The complete perching maneuver occurs in a fraction of a second, within approximately 200 ms (see Fig. 2a). It begins with the UAV flying directly toward a pole at a certain speed and angle of attack to make a primary impact with the nose. The impact energy causes the robot to start rotating in the pitch direction and release its pre-loaded wings. The maneuver concludes with a secondary impact on the fuselage or tail to halt the rotation and the wings hugging the pole to hold the UAV in place. Only the correct sequence of events can lead to a successful landing on the pole.

The UAV design consists of a nose and wings design that can serve the dual purpose of flight and crash-perching (Fig. 2b-g). This design strategy does not require additional hardware such as dedicated perching claws or feet to successfully perch the UAV. We present the principles behind the design of the integrated hardware of PercHug, encapsulated in an Expanded PolyPropylene (EPP) foam body, with a ready-to-perch weight of 550 g and a wingspan of 96 cm (Fig. 2; see Methods for the fabrication details).

The design of the upturned nose shape (Fig. 2b) enables the robot to passively reorient from horizontal flight to the vertical configuration needed for perching. The primary impact force at the tip of the nose, which is large in magnitude, generates a moment about the center of gravity (COG) to yield rotation, implying that the relative placement of the nose tip with respect to the COG plays a crucial role in reorientation performance (Fig. 2c). In addition to the upturned nose shape, we also studied the reorientation performance by incorporating nose extensions using a flexible flat carbon bar (Fig. 2b and c).

The vehicle is equipped with foldable wings that have three hinged segments. One segment is attached to the fuselage, while the other two can bend in the ventral direction to wrap around the pole (Fig. 2d). Three torsion springs, in parallel configuration with a combined stiffness of 3.45

N · mm.$^{\circ-1}$, are placed at the interface between the two segments and are pre-loaded during flight. Upon impact, these springs are released and cause the segments to fold and press against the pole. This gripping force, combined with the friction along the vertical axis, keeps the robot attached to the pole.

The outermost segments of the wings can be equipped with nine removable hooks (Fig. 2e) to help engage with rough surfaces (such as the bark of a tree). A tensioning wire keeps the wings open and straight during flight. The wire connects the two tips of the wings to a latch in the fuselage (Fig. 2b), and its length is adjusted to provide a dihedral angle of about 5° in order to improve the lateral stability of the aircraft.

The latching mechanism (Fig. 2f) holds the wire in tension during flight and passively releases it upon impact. It comprises a fixed piece (shown in blue) firmly attached to the fuselage and a latch (shown in red) through which the wire passes. The pulling force exerted by the wing springs on the wire, connected to the latch, is blocked by the vertical wall of the fixed blue piece during flight. Wings are released when the latch pops over the blocking wall. We can adapt the release time by adjusting the wall height. With a height of 5 mm, the latch is released upon primary impact with the pole (Fig. 2a-2) due to the shock taken by the airframe. If the walls are 10 mm high, the wings are not released at primary impact but are unlatched at the secondary impact once fully reoriented vertically (Fig. 2a-4). This is made possible with the backup bistable trigger (Fig. 2g), which operates similarly to the mechanism in[35]. It sits right beneath the latch and connects to a switching pad that extends out of the fuselage during flight, aided by a pair of compressed springs. A secondary impact with the underside of the fuselage triggers the release passively. The impact pushes the pad inward, causing the bistable mechanism to switch its position. As a result, the pushing rod moves upward and releases the latch by popping it over the blocking wall, freeing the wings.

### Inertial reorientation

We decoupled the reorientation and pole-hugging problems to independently study nose selection and wing design. Here, we primarily investigate how the upturned nose, with and without elastic nose extensions, performs in reorienting the robot at impact (Fig. 3). The rationale behind using flexible noses is to examine whether they enhance reorientation by extending the impact moment arm with respect to the COG, decreasing the maximum impact force taken by the airframe, and redirecting the force vector more effectively toward sliding up and attaching to the surface.

We consider a reorientation successful if the UAV reaches a vertical orientation and makes secondary contact with the wall (Fig. 3a). In contrast, a failure means a rebound off the surface after the primary impact and a lack of secondary impact. The tracking data for a successful reorientation case is illustrated in Fig. 3b. The primary impact phase (highlighted in red) indicates the start of the reorientation and is identified by the sharp drop in translational speed and the sharp increase in pitch rate. The end of the reorientation maneuver is also considered to be the point of maximum pitch angle (marked by the blue line). The impact speed $V_i$ and relative impact

**Fig. 1 | Avian-inspired utilization of forelimbs for perching on trees.** A straw-coloured fruit bat (*Eidolon helvum*) holding onto a tree branch using its wings and clawed feet (left), a great grey owl (*strix nebulosa*) fledging on its first day out of the nest wrapping its wings around a tree trunk to rest during climbing (center), and the PercHug robot perching vertically on a tree by hugging (right). Photo credits[46,47].

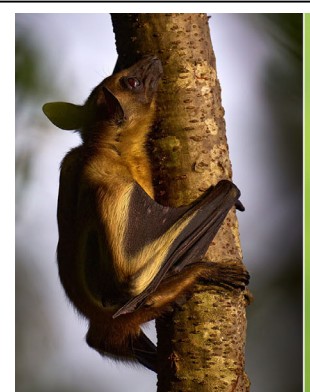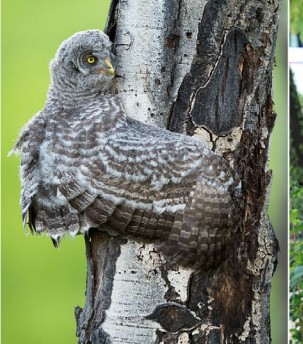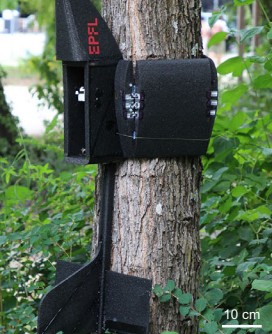

10 cm

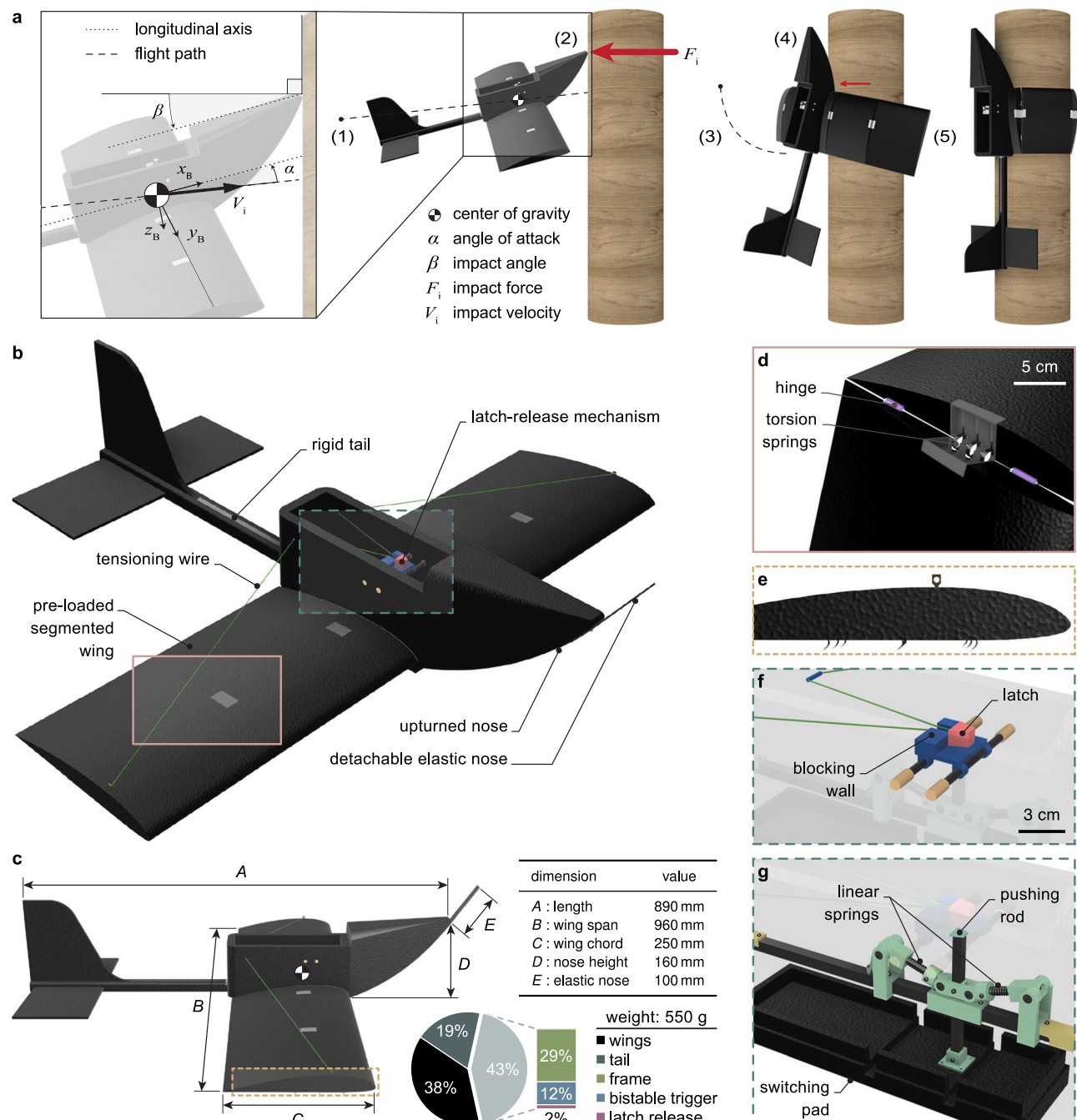

**Fig. 2 | Perching strategy and architecture of the PercHug platform. a** Operating principle of PercHug depicting the key steps of the perching maneuver: (1) gliding, (2) primary impact, (3) reorientation and wing release, (4) secondary impact, and (5) wing-wrapping. The red arrows represent the expected magnitudes of the impact forces, proportionally drawn. **b** Isometric view of PercHug showing different elements of the robotic platform. **c** Side view and physical properties of the robot. **d** Pre-loaded segmented wing interface in an open configuration. **e** Side view of the outermost wing segment highlighting the hooks. **f** Latching wing release mechanism (blue and red). **g** Backup bistable trigger (green).

angle $\beta$ (shown in Fig. 2a) are estimated from the moment of primary impact. Notably, $\beta$ corresponds to the robot's pitch at impact, given that the wall is placed vertically.

The reorientation success rate depends primarily on the impact angle rather than on variations in impact speed. This behavior is clearly seen in Fig. 3c, where reorientations fail below a certain impact angle threshold, regardless of the type of nose used. With the standard upturned nose, the vehicle successfully reorients for impact angles above about 15° at speeds of 3 m · s⁻¹ to 9 m · s⁻¹. Such speeds match cruising speeds of comparable vehicles in size and weight, such as ref. 6. In comparison, reorientation performance improves with increased flexural

rigidity $D$ of the elastic noses (see Methods for the definition of flexural rigidity). This trend is seen by the successful reorientations at lower impact angles, reaching up to a minimum of 8° for the stiffest nose. The results thus indicate that the elastic nose extension with $D = 0.233\,\text{N} \cdot \text{m}^2$ is the best-performing one out of the nose types tested. Despite the differences among the success rates, the duration of the reorientation maneuver remains consistent for different speeds, angles, and nose types. We define this duration as the time from primary impact to 90° pitch and measure it as an average of $196 \pm 59$ ms.

Unlike the success rate, the primary impact force $F_i$ (shown in Fig. 2a), which is estimated from the speed profile, is linearly proportional to the

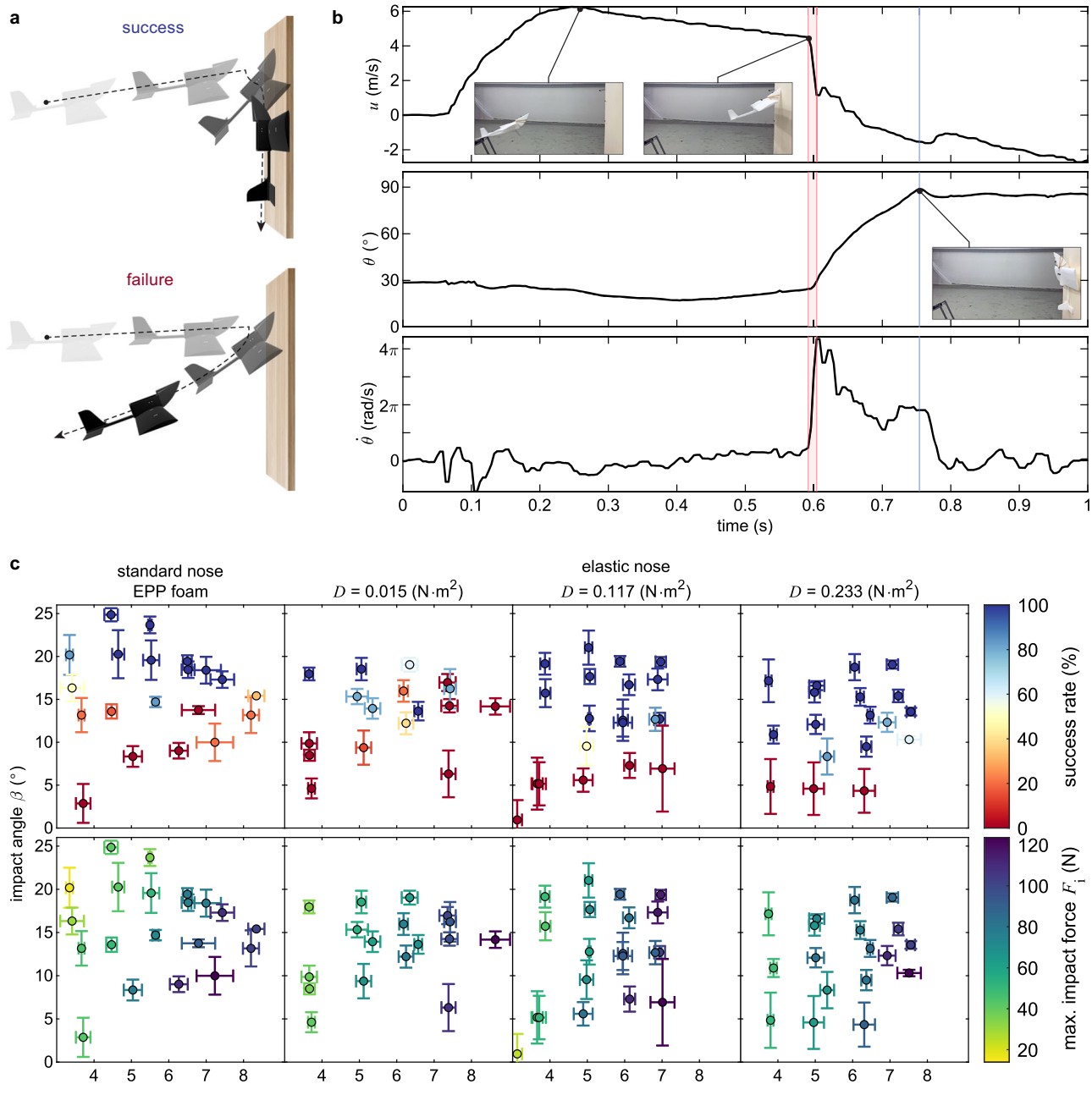

**Fig. 3 | Reorientation maneuver and performance with four different noses.**
**a** Illustration of the concepts of unsuccessful reorientation, where the Unmanned Aerial Vehicle (UAV) bounces off the wall after impact, and a successful one in which it reaches a vertical orientation while making secondary contact with the wall. **b** Time evolution of the UAV's translational velocity $u$, pitch angle $\theta$, and pitch rate $\dot{\theta}$ for a sample trial (see Supplementary Fig. S2 and Methods for the definitions of the state variables). The red region shows the duration of the primary impact, and the blue line corresponds to the time of maximum pitch for a successful reorientation. **c** Characterization results of the UAV reorienting from horizontal to vertical configuration. The plots show variations in success rate and maximum primary impact force with impact angle and speed for four different types of noses. The error bars represent one standard deviation of repeated experiments.

impact speed $V_i$ (see Methods for mathematical formulation). Its peak value varies between about 15 N to 120 N for speeds of 3 m·s⁻¹ to 9 m·s⁻¹ (Fig. 3c). It is also expected that there will be a linear correlation between weight and impact force (Eq. (6)). In the case of the standard upturned nose, the primary impact force also exhibits a marginal dependence on the impact angle, with slight increases observed as the impact angle decreases. However, the correlation is unclear for the other nose types possibly due to the effect of the nose flexibility. Although the elastic nose with $D = 0.233\,\text{N}\cdot\text{m}^2$ improves the success rate at lower impact angles compared to the standard upturned nose, Fig. 3c also shows that they share similar amounts of impact force over the range of tested impact speeds. The observed improvement is associated with the elongated moment arm between the impact point and COG (Fig. 2a and Supplementary Fig. S2). According to the simplified reorientation dynamics (Eq. (9)), pitching acceleration is influenced by the rigid nose tip offset parameters. This effect is notable in the stiffest elastic nose extension that acts more like a rigid nose and transmits a greater force and moment to the COG, which causes a faster pitching acceleration (Fig. 3c). Moreover, Eq. (9) suggests that at lower pitch angles the vertical nose offset contributes more to the pitching moment, while the longitudinal offset has minimal effect. Hence, enhancing success at lower impact angles can be accomplished by increasing the vertical nose tip offset. The data presented aids in estimating impact forces for similar-sized robots at varying

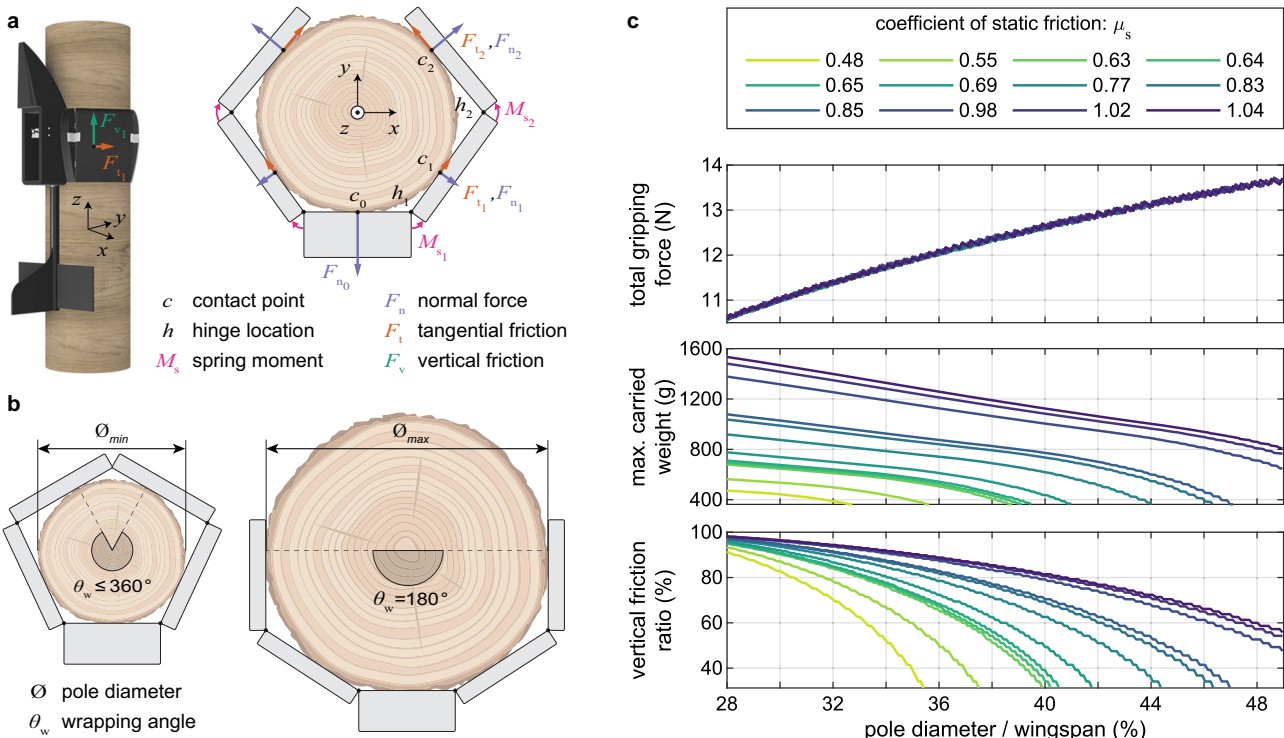

**Fig. 4 | Static wing-wrapping model and pole gripping performance. a** Free-body diagram used for static modeling of the robot perched on a pole, shown in isometric and top views (refer to Methods for further details). **b** The practical limits of the pole diameters the robot can perch on. **c** Simulation results of the static model showing variations in net squeezing force by the wings, maximum static payload capacity, and friction split with pole size and material. The static friction coefficients correspond to the poles used for the actual static experiments, while the diameter range is defined by the minimum and maximum values.

weights, offering valuable insights for airframe mechanical design and structural analysis.

## Static perching

The sizing and segmentation of the folding wings define the range of pole diameters on which the robot can perch. In order to understand the most suitable dimensions, we developed a wing-wrapping model (Fig. 4a; see Supplementary Fig. S3 and Methods for details), and validated it by conducting pole-hugging experiments with the predicted wing design. The model applies to static perching with hook-less wings, referring to the situation when the wings are already wrapped around the vertical pole.

In contrast to insects, which use dry or wet adhesion or claws to hold onto tree surfaces, larger animals with articulated limbs leverage interlocking methods by encircling over half the trunk with their forelimbs (see Fig. 1)[36]. Similarly, here we assume that the maximum pole diameter on which the UAV can practically perch corresponds to a wing-wrapping angle ($\theta_w$) of 180°, and the minimum pole diameter corresponds to a size that prevents the two wingtips from overlapping, (Fig. 4b). This wing-wrapping angle is defined by the portion of the pole covered by the folded wings, between the two contact locations along the outermost segments. Its maximum value depends on the wingspan and the number and lengths of the segments, approximately reaching 290° for our selected segmentation size. For the wings to remain wrapped, the normal and tangential reaction forces of the outermost segments ($F_{n_2}$, $F_{t_2}$), which have components directed in +$y$, must be sufficiently large to counteract the −$y$ directed forces on the fuselage and other segments. In the event that a pole of larger diameter is employed ($\theta_w < 180°$), it follows that almost no force exerts in the +$y$ direction, which inevitably causes the robot to slide off the pole, unless for the very unlikely cases of high friction coefficients.

The model takes into account various body and wing characteristics, including wingspan, fuselage width, the number and sizes of the folding-wing segments, and the stiffness of torsion springs, along with the coefficient of static friction between EPP and the pole material. These factors help

estimate the range of diameters on which the UAV can perch and the maximum static payload. The range of pole diameters suitable for perching mainly scales with the wingspan and spans from approximately 28% to 50% of it, as predicted by the model. For the selected 960 mm wingspan with two folding segments per wing, this corresponds to diameters of 265 mm to 470 mm.

The model was also used to study variation in gripping force and maximum static payload with the pole diameter Ø and coefficient of static friction $\mu_s$. These two factors critically influence the wing-wrapping performance. The analysis revealed a positive correlation between the net gripping force and the pole diameter (Fig. 4c). This can be attributed to the greater compression of the torsion springs between the wing segments, leading to a stronger squeezing force on a wider pole. However, the maximum weight the UAV can support before sliding down the pole reduces as the diameter increases. This is due to the sharp decrease in the fraction of the total friction force acting along the vertical axis, which outweighs the increase in pressing force on wider diameters. As a result, although the total friction force increases with an increase in pole diameter, its vertical component ($F_v$) decreases, limiting the amount of mass the UAV can hold. Concerning the effect of the coefficient of static friction, an increase in $\mu_s$ leads to an improvement in overall static perching performance, as expected.

We utilized the wing-wrapping model to determine suitable dimensions for the segments and validated the selected wing design with experiments of a physical prototype attached to poles of different sizes and materials. The surface texture and specifications of the tested poles are listed in Fig. 5a and b. Our symmetrical wing segmentation approach yielded folding segments of size 195 mm (see Supplementary Fig. S4 and Methods for the segmentation study). The results exhibit a strong agreement between the measured values and the model predictions (Fig. 5c). As predicted by the model, the experiments show that the maximum payload increases as the diameter decreases, assuming the surface material remains the same (evidenced by pairs I-II, VI-VII, and VIII-IX). Despite the variation in diameter, an overall upward trend in the payload is observed from left to right,

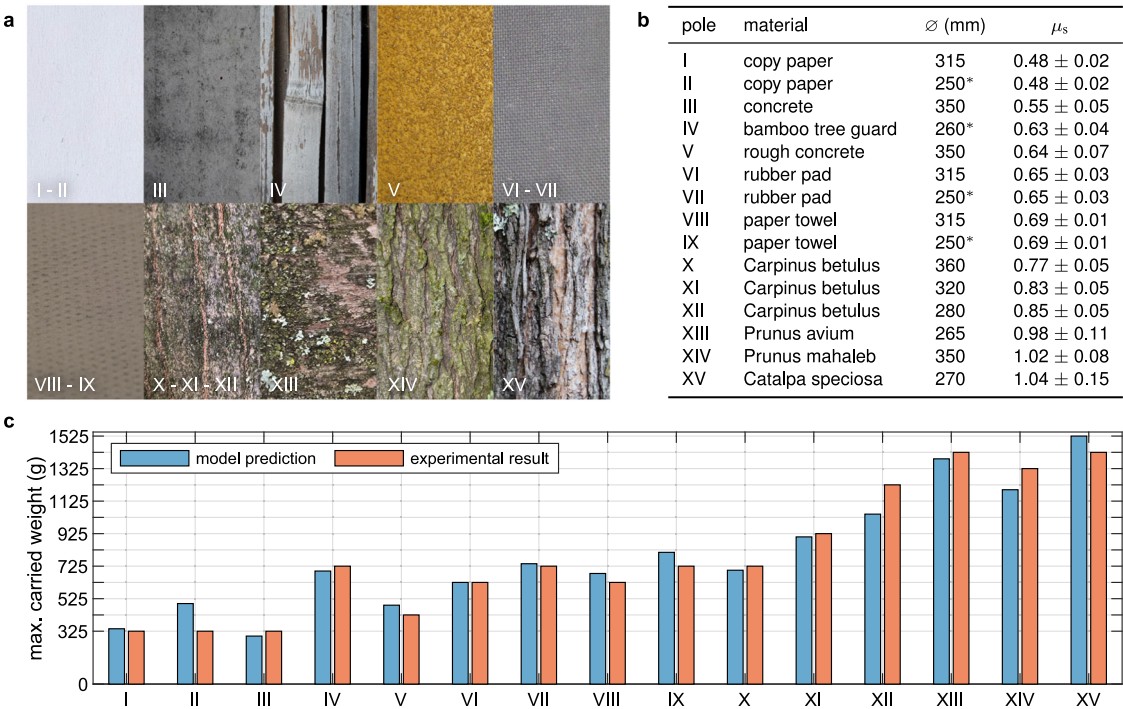

**Fig. 5 | Model validation with static perching experiments. a** Close-up pictures of the surfaces of the poles used in the static perching experiments. **b** List of poles and their specifications in order of increasing friction coefficient. The "*" symbol denotes poles with a diameter smaller than the model's predicted minimum value of 265 mm. These cases were analyzed since the model was found to be valid even outside the previously mentioned diameter range, provided that the considered diameter is close to the limit. **c** Comparison of model predictions and actual measurements for the maximum static payload capacity. The estimated payload started with measurements at the weight of the prototype alone (325 g) and incremented by 100 g steps until failure (see Methods for more details). The insignificant discrepancies between experiment and simulation results in cases II, XII, and XIV can be attributed to minor errors due to the non-uniformity of tree barks (XII and XIV) and possible discrepancy of the stiffness coefficient of the torsion springs for small poles (case II) that exceed the manufacturer values for linear operation.

highlighting the positive effect of increasing the coefficient of friction on payload capacity.

## Experimental validation with PercHug

We leveraged the insights gained from the reorientation studies and the static perching model to size and characterize the hug-perching performance of PercHug when hand-launched against trees. PercHug weighed 550 g (see Fig. 2c for the weight distribution), including the unlatching mechanism, bistable backup trigger, and folding wings with hooks (Fig. 2d–g), as well as a reinforced tail and body to enhance durability during multiple crash-perching tests. In these experiments, we tested PercHug equipped with and without the extended elastic nose of $D = 0.233 \, \mathrm{N \cdot m^2}$ (corresponding to the best reorientation performance) on the six trees used in the static perching experiments (trees X-XV in Fig. 5).

In all dynamic perching experiments, we hand-launched the robot toward the trees. A perching trial was considered successful if it involved four distinct phases of gliding, reorienting at impact, wrapping the wings, and staying perched on the tree (see Fig. 6a and Supplementary Video S1). The efficacy of hooks and the critical role of unlatching time in increasing perching success became evident during our preliminary investigations (Supplementary Video S2). Unlike in static perching, the robot often encountered the issue of sliding down the pole with wings wrapped and a challenge to come to rest. The incorporation of hooks mitigated this problem significantly and was observed to be effective in over one-third of the successful trials. Hooks, through their randomized engagement with surface asperities, facilitated a rapid deceleration, effectively bringing the robot to a swift stop. We also investigated the effect of wing release timing by adapting the latch mechanism to release at either the primary or secondary impact using the bistable trigger (Fig. 2a–g). Experiments revealed that the secondary impact release strategy resulted in nearly always unsuccessful trials,

regardless of the nose type. This was due to the robot falling off the tree shortly after the secondary impact, rather than just falling straight down, and as a result, it was pushed away from the tree making it almost impossible for the wings to wrap around. Conversely, when using the primary impact release strategy, the robot displayed more robust perching capabilities.

Analyzing the video and tracking data from successful perching trials (see Methods for further information) reveals that the release mechanism was triggered very shortly after the primary impact, with about a 25 ms delay (Fig. 6a, b). Additionally, the wrapping phase concluded almost simultaneously with the end of the reorientation phase, which is characterized by the instance of reaching an approximately 90° pitch. In this specific test, the durations of reorientation and wrapping, measured from the moment of impact, were approximately 150 ms and 180 ms, respectively. These results confirm the rapid dynamics of the perching maneuver and underscore the importance of timing the unlatching strategy to release at the primary impact.

Perching success is also significantly influenced by the quality of approach conditions. Impacting the target pole with the nose at its centerline emerges as a critical factor, as deviations in lateral direction or angular errors can lead to failure (Supplementary Video S3). Substantial lateral offset risks impacting the tree with the wings rather than the nose, resulting in no reorientation, and hence, failure. Moreover, even if the impact is on the nose but off-centered, there is a potential scenario where one wing wraps around the trunk while the other remains too distant, leading to unsuccessful perching. Angular errors in all degrees of freedom can also contribute to failure. As expected, a low impact angle, which generally resembles a low pitch angle, leads to insufficient reorientation and a rebound off the tree. Furthermore, errors in roll and yaw can lead to a missing tail impact and subsequent failure. The rigid tail of PercHug plays a crucial role in halting the reorientation maneuver upon contact with the trunk when reaching a vertical position. This is evident from the pitch data, where the pitch increase

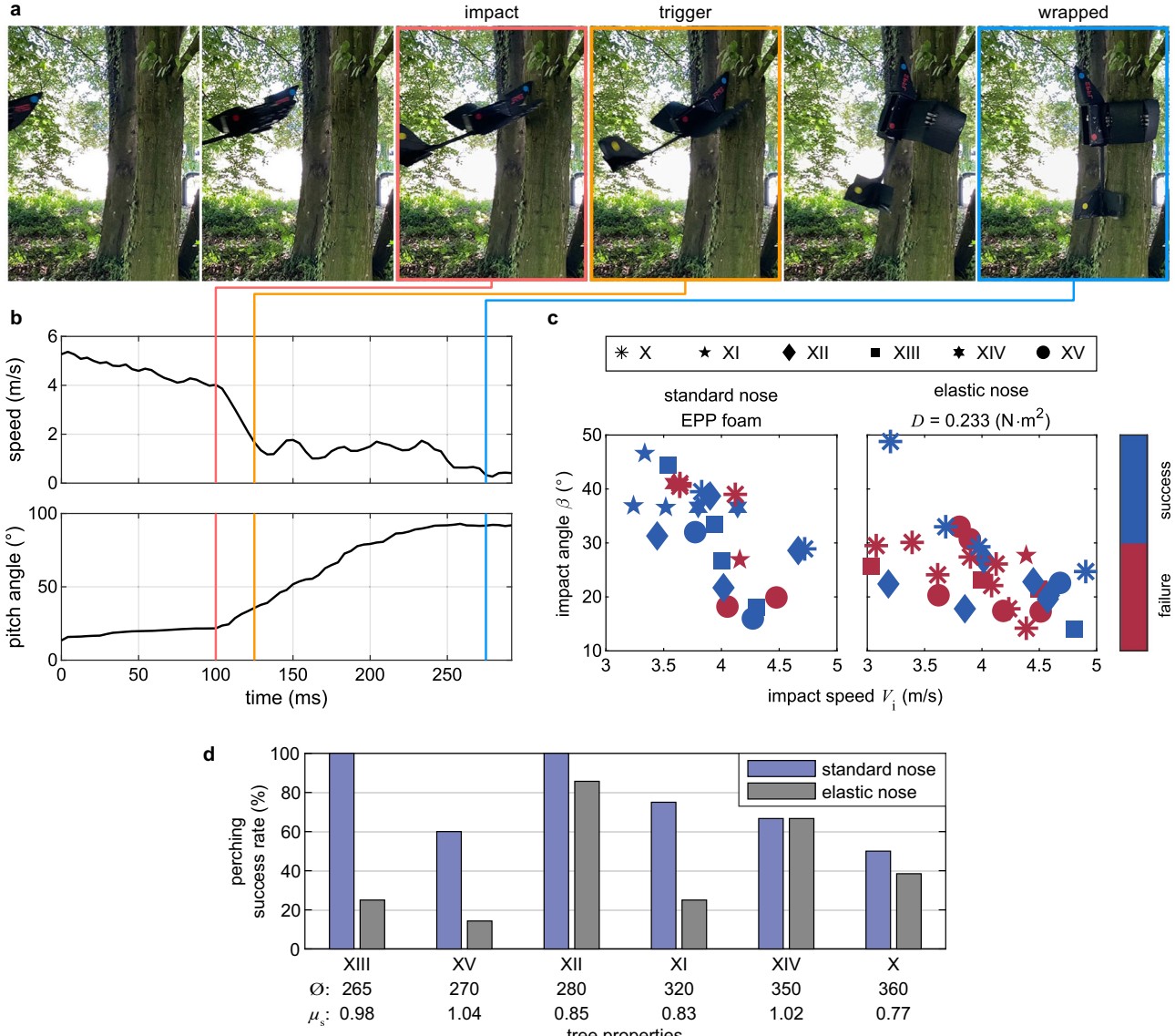

**Fig. 6 | Dynamic perching performance on trees with two different nose configurations. a** Snapshots of a crash-perching experiment with PercHug on a tree trunk (XII), captured from high-speed footage (Supplementary Video S1). **b** The speed and pitch angle variation during the dynamic perching maneuver (see Methods for further information). PercHug glides and settles at a speed of around 4 m · s⁻¹ with a pitch of 20° before impact with the tree. The sharp decrease in speed and increase in the pitch angle indicates the beginning of the reorientation phase.

Triggering occurs right after the primary impact and the tensioning wire is released. The wings are seen wrapping and closing around the tree during reorientation. The wrapping phase concludes when the wingtips make contact with the rear surface of the tree trunk. **c** Success map of dynamic perching on various trees with the standard upturned nose and extended elastic nose. **d** Perching success rates on each tree, ordered by increasing diameter.

ceases and levels off at around 90° due to the tail's contact (Fig. 6b). In trials where the robot did not make contact with its tail, it continued to reorient beyond 90°, resulting in unsuccessful landing maneuvers. This finding aligns with the research conducted by Siddal et al.[29] on the impact of tails in tree-perching geckos, which emphasized the advantages of longer-tailed robots when crash-landing at speeds between 3 m · s⁻¹ to 5 m · s⁻¹.

PercHug successfully demonstrated crash-perching capability on all trees for impact speeds $V_i$ ranging from 3 m · s⁻¹ to 5 m · s⁻¹ and relative impact angles $\beta$ above 15°, regardless of the nose type (Fig. 6c). The success maps illustrate the distribution of successful and failed perching trials across different trees. These experiments occurred to have very similar average impact speeds of 4.1(0.7) m · s⁻¹ for both nose configurations, indicating a notable consistency across the conducted trials despite hand-launching.

Perching was more successful with the standard upturned nose compared to the elastic nose on the majority of tested trees (Fig. 6d). The standard nose configuration yielded an overall perching success rate of 73%

across all trials, significantly outperforming the 42% achieved with the elastic nose (see Supplementary Table S1 for a detailed breakdown). While the elastic nose improved reorientation by facilitating the nose slide-up on the pole (Fig. 3c), it hindered surface attachment by creating a larger gap between the robot and the tree, making it harder for PercHug to wrap around. In other words, some of the kinetic energy was stored as elastic energy inside the bent nose and released, effectively pushing the robot away from the tree. The standard nose configuration exhibited a 30% faster triggering time compared to the experiments with the elastic nose, which potentially contributed to its higher success rate. The average trigger delays from impact were 26(7) ms and 37(15) ms, for the standard and elastic noses, respectively. In successful trials, the respective mean wrapping times were 156(30) ms and 152(25) ms, with a variation of less than 3% for the standard and elastic noses (see Supplementary Table S1). The comparable wrapping times imply that this parameter is likely a characteristic of the wings and unaffected by other UAV components.

With the exception of tree XV, experimental results with the standard nose also indicate that wider trees lead to lower success rates (Fig. 6d). This behavior aligns well with the trend predicted by the model (Fig. 4c) and observed in static perching experiments. While the coefficient of static friction $\mu_s$ and pole diameter Ø are equally crucial factors for static perching, the diameter of the tree plays a more significant role in dynamic perching success, provided that the friction is sufficient for the robot to maintain its grip.

## Discussion

We have presented a bio-inspired method to passively crash-land on vertical poles and trees with winged robots. While flying animals substantially reduce their kinetic energy at landing by wing flapping, specialized gliders, like flying squirrels and geckos, land on trees at high speeds and endure elevated forces utilizing their limbs or head[37]. We took inspiration from flying geckos, which exhibit head-first crash-landing at speeds between 5 $m \cdot s^{-1}$ and 7 $m \cdot s^{-1}$ [29]. In a similar manner, our proposed upturned nose robustly transforms the impact kinetic energy to a passive reorientation maneuver at 3 $m \cdot s^{-1}$ to 9 $m \cdot s^{-1}$, thus eliminating the need for a complex pitch-up maneuver to bleed off speed before perching. Additionally, geckos have been observed to reach the target with their body pitched upward between 8° and 24° [29]. Our experimental results align with these findings, showing that a minimum impact angle of 15° with the upturned nose (or 8° with an extended elastic nose) is crucial for successful reorientation. Although these angles are small, optimizing the position of the nose tip relative to the COG can enable flight and reorientation at smaller angles of attack. The impact force, speed, and angle characterization results, coupled with the robot's weight and inertia data, provide the foundation for developing a dynamic model to further explore design possibilities at different speeds, angles of attack, and pole dimensions.

Large-sized mammals lacking sharp claws, and less commonly some flying animals (Fig. 1), use the interlock method when climbing or resting on tree trunks by encircling at least half of the trunk with their forelimbs[36]. This behavior inspired the design of our foldable, pre-loaded segmented wings that serve the dual purposes of flight and perching by securely wrapping around poles after impact. Our static perching model and tests similarly showed that we are constrained by a wrapping angle of 180°, and that the range of pole sizes the robot can perch on is solely determined by the choice of wingspan and segmentation. Moreover, the pole diameter and the coefficient of static friction are discovered to be equally important factors for maintaining a secure perch. The results highlighted that the static payload capacity varies inversely with the diameter and directly with friction. These findings are in line with those observed in nature reporting the interlock fastening method's dependency on the angle of the frictional force, which leads to a reduced gripping force when animals climb larger branches[36,38]. We also investigated the sizing relationship and segmentation of the wings through our model, providing insights that can aid in the design of improved wings and the prediction of performance for robots of different dimensions.

PercHug's experimental results have brought us invaluable insights into dynamic perching on trees. Two crucial factors have been identified as determinants of perching success. First, the rapid dynamics of the perching maneuver, occurring in under 200 ms, underscore the importance of precise timing for wing release. Optimal results are achieved by releasing the wings at the beginning of the reorientation phase upon primary impact while releasing them at the end of reorientation by the secondary impact leads to almost complete perching failure. Second, tree diameter has emerged as a more dominant factor than the friction coefficient in determining the success rate. Consistent with the predictions of the static model, perching improves on smaller tree sizes within the acceptable range of diameters compared to larger ones. It is also worth noting that despite the elastic nose's potential for improved reorientation, its detrimental push-away effect resulted in a lower perching success rate (42%) compared to the standard upturned nose (73%).

While we successfully demonstrated a fully passive pole perching method, our platform currently lacks flight and targeting capabilities prior to perching, along with the ability to unperch and take flight again. To address these shortcomings, future developments based on this work will integrate avionics and proper control surfaces to enable autonomous flight and perching. Ongoing efforts involve sensor-based pole detection for perching, a powered grip loosening control for thrust-assisted climbing, and unperching, collectively contributing to a complete mission cycle. In the planning phase of a perching mission, a GNSS receiver can provide means to autonomously navigate from far away to a target region within a few meters of accuracy[39]. Subsequently, a target pole can be detected using vision-based methods from tens or hundreds of meters away[40–42]. Switching to an accurate vision-based flight control method can then facilitate maneuvering to the desired spot, as demonstrated in ref. 43. The success or failure of the perching maneuver can be determined using data from an inertial measurement unit (IMU). Additionally, the propulsion scheme is a critical consideration, favoring a rear fuselage-mounted one due to the impracticality in mounting propellers on the nose or wings. The incorporation of a propulsion system also holds the potential for further enhancing perching success by utilizing the thrust to push toward the pole and against gravity to reduce the bounce-off effect during the landing maneuver. Given the pre-loaded and hugging design of the wings, a crucial enhancement to the system involves transforming the latch-release mechanism into a reversible powered subsystem, enabling active control over wing opening and closing. This modification unlocks a range of new functionalities. First, it can facilitate thrust-assisted climbing by slightly opening the wings to loosen the gripping force, allowing movement across the pole. A quick-release capability enables a seamless return to perching mode upon reaching the target spot on the pole. Second, it facilitates full wing opening for unperching, and third, aiding recovery from a failed perching event identified by the IMU to prevent a fall to the ground. Executing a controlled recovery maneuver involves reopening the wings quickly, coupled with using thrust and adequate control authority from aerodynamic surfaces to regain stable flight. Overcoming the high spring forces at the wing joints presents a challenge in designing this active wing opening mechanism, which necessitates using a high-torque motor, which can be slow and heavy, to pull onto the tensioning wires given their short moment arm with the springs. Extending this arm and guiding the wire across higher points on the wings can alleviate this issue. Additionally, the slower wing opening rate compared to wing closing requires a solution for remaining attached to the pole during unperching. Potential strategies involve utilizing thrust for momentary hovering, incorporating retractable hooks under the fuselage for engagement, or utilizing materials with bidirectional friction properties to facilitate upward movement while preventing downward falls. These advancements will mark a pivotal step in overcoming existing limitations, contributing substantially to the evolution of a more comprehensive aerial system.

We firmly believe that our study lays a foundation for advancing perching technologies and paves the way for the development of highly versatile robotic systems tailored to diverse applications. Such robots could be deployed for inspection tasks in complex industrial environments or tall buildings, enabling close-up examinations without the need for scaffolding or risky human interventions. In the field of infrastructure inspection and maintenance, perching would enable them to access challenging locations with ease, like electric or cellular towers, allowing for the inspection of power grid systems and communication framework by assessing equipment functionality. Perching on lamp posts or street signs in urban environments could enhance surveillance and security systems. In environmental monitoring applications, these versatile robots could perch on trees to gather data on biodiversity, habitat conditions, and ecological changes. Additionally, they could serve as valuable tools for studying wildlife behavior, enabling non-intrusive data collection to support wildlife conservation efforts. The possibilities are vast, and as perching solutions evolve, these robotic systems will find their place in numerous domains, reshaping the way we interact with and benefit from intelligent aerial machines.

## Methods
### Robot fabrication
The PercHug robot (shown in Fig. 2) is made of three main material groups: Expanded Polypropylene (EPP) flexible lightweight foam, 3D-printed

Tough Polylactic Acid (PLA), and fiber-reinforced carbon bars. The upturned nose, wings, tail, and soft shells of the fuselage are made out of multiple patterns cut out of EPP using a hot wire foam cutting tool. Each element is created separately and then assembled at the final stage.

The fuselage shells are glued together with UHU por and encapsulate the internal skeleton made out of an 8 mm x 8 mm pultruded carbon fiber square tube. The pieces of the latching wing release mechanism, which sits over two 4 mm round carbon tubes at the top of the fuselage, as well as the backup bistable trigger, are 3D-printed out of Tough PLA. The bistable trigger's pushing rod (4 mm x 4 mm square carbon bar) connects to the bottom shell of the fuselage (switching pad). The rest of the system assemble through connecting the pieces along with two linear springs and eight bearings, whose interfaces are directly glued to the fuselage's carbon bar.

Three identical tough PLA 3D-printed pieces hold M2 threaded brass inserts, which are inserted and glued with five-minute Epoxy to the underneath of the foam nose for the attachment of the elastic extensions. The foam nose is then directly glued with UHU por to the front shell of the fuselage. The tail is reinforced with a 6 mm x 6 mm pultruded carbon fiber square tube, to which its foam pieces are glued with five-minute Epoxy. The square tube is tightly fitted into the main fuselage carbon bar, making the tail replaceable in case of damage. Each wing is cut into corresponding segment sizes, and every two segments are held together with two 20 mm x 36 mm plastic hinges, glued to their flat bottom side. A square channel is cut at every segment's interface, where the housing for the three torsion springs is fitted and glued. We use five-minute Epoxy to interconnect the wings parts. Additionally, nine fishing hooks are hot-glued to the bottom side of each outermost segment. The left and right wings are brought together and their fixed segments are assembled to the fuselage using two cross-round carbon tubes with a diameter of 5 mm. Lastly, two eye screws are fixed to 3D printed pieces on the wing tips through which we pass the composite fiber (Dyneema) cords, with a thickness of 0.2 mm and a maximum load capacity of 20 kg, that connect to the latch in the fuselage.

## Reorientation experiments

The experimental setup shown in Supplementary Fig. S1 was built to characterize the reorientation performance based on impact speed $V_i$, impact angle $\beta$, and nose type (see Fig. 2). We tested four different noses, one with the standard upturned foam shape and three with extended elastic beams of different cantilever flexural rigidity $D$ (see Elastic noses), to test the effectiveness of the flexible noses. We used a 220 g fixed-wing glider, made entirely out of EPP foam with dimensions given in Fig. 2c, and equipped it solely with these nose types to eliminate potential effects due to other UAV elements. We launched the robot with a bungee-powered catapult with adjustable speed and angle toward a vertical wall. For each nose, we tested the vehicle at speeds of approximately $3\,\mathrm{m\cdot s^{-1}}$ to $9\,\mathrm{m\cdot s^{-1}}$ and positive pitch angles of up to 25°. We conducted a total of more than 100 experiments, each repeated with at least five trials, tracked the robot's position and attitude in space by an OptiTrack motion capture system (see Flight kinematics and impact force estimation for the derivation of other states), and recorded slow-motion videos with a camera at 240 Hz (Supplementary Fig. S1).

## Elastic noses

Flexural rigidity or bending stiffness of a cantilever beam, which is fixed at one end and free at the other, is the measure of its resistance to bending subject to an external load. It is measured as the product of the modulus of elasticity $E$ and area moment of inertia $I$ of the beam. For the elastic noses, we used flat pultruded carbon fiber profiles with rectangular cross-sections, whose flexural rigidity is calculated by

$$D = EI = \frac{Ebh^3}{12}, \qquad (1)$$

where $b$ and $h$ are the width and height of the cross-section, respectively. A higher flexural rigidity indicates a stiffer beam, which can withstand larger bending loads and experience smaller deflections.

## Flight kinematics and impact force estimation

Here, we present the mathematical expressions used to estimate the robot's state variables and impact parameters. These estimations rely on the tracking data obtained from the motion capture system (OptiTrack) during the reorientation experiments. In our notation, subscripts $B$ and $I$ denote variables in the body-attached and inertial reference frames, respectively, and subscript $i$ designates the magnitude of parameters at primary impact.

We assume the robot is a rigid body, defined in space through three positions and three linear velocity states for translational motion, along with three angular positions and three angular velocity states for rotational motion (Supplementary Fig. S2). The motion capture system is only capable of tracking the position and attitude angles over time. $\overrightarrow{\mathbf{P}}_I = [\,x \quad y \quad z\,]^\top$ denotes the position of the body-attached frame, placed at the robot's COG, with respect to the inertial reference frame. $\overrightarrow{\Gamma} = [\,\phi \quad \theta \quad \psi\,]^\top$ defines the body orientation with attitude angles (roll $\phi$, pitch $\theta$, and yaw $\psi$) according to the Euler angle convention[44]. The linear velocities $\overrightarrow{V}_B = [\,u \quad v \quad w\,]^\top$ and angular velocities $\overrightarrow{\Omega}_B = [\,p \quad q \quad r\,]^\top$ of the vehicle, along the body-attached axes, are estimated by

$$\overrightarrow{\mathbf{V}}_B = \mathbf{R}_{BI}\,\dot{\overrightarrow{\mathbf{P}}}_I, \qquad (2)$$

$$\overrightarrow{\Omega}_B = \mathbf{J}\,\dot{\overrightarrow{\Gamma}}, \qquad (3)$$

where the rotational transformation matrices are

$$\mathbf{R}_{BI} = \begin{bmatrix} c\theta c\psi & c\theta s\psi & -s\theta \\ s\theta s\phi c\psi - s\psi c\phi & s\theta s\phi s\psi + c\psi c\phi & c\theta s\phi \\ s\theta c\phi c\psi + s\psi s\phi & s\theta c\phi s\psi - c\psi s\phi & c\theta c\phi \end{bmatrix}, \qquad (4)$$

$$\mathbf{J} = \begin{bmatrix} 1 & 0 & -s\theta \\ 0 & c\phi & s\phi c\theta \\ 0 & -s\phi & c\phi c\theta \end{bmatrix}, \qquad (5)$$

with c and s being shorthand notation for cos and sin, respectively. The derivative of the robot position $\overrightarrow{\mathbf{P}}_I$ ($\overrightarrow{\mathbf{V}}_I$) and attitude angles $\overrightarrow{\Gamma}$ are numerically calculated from the tracked data using the central difference method.

For each reorientation experiment, we determine the primary impact phase (highlighted in red in Fig. 3b) based on the abrupt change in the state values of translational velocity $u$ and angular velocity $q$ (almost equivalent to the pitch rate $\dot{\theta}$, according to Eqs. (3) and (5), when the roll angle $\phi$ is nearly zero). For successful trials, the end of the reorientation maneuver, or approximately the onset of the secondary impact, is characterized by reaching the maximum pitch angle $\theta$ (almost 90°). We estimate the maximum primary impact force, denoted as $F_i$, by analyzing the change in acceleration profile during the primary impact. If $m$ denotes the total mass of the vehicle and $a_i$ is the magnitude of the maximum acceleration data point within the primary impact phase, then

$$F_i = m a_i. \qquad (6)$$

The acceleration can be estimated by numerically differentiating the speed profile using the central difference scheme, either from the inertial velocity $\overrightarrow{\mathbf{a}}_I = \dot{\overrightarrow{\mathbf{v}}}_I$ or the body-fixed variables

$$\begin{aligned} \overrightarrow{\mathbf{a}}_B &= \dot{\overrightarrow{\mathbf{v}}}_B + \overrightarrow{\Omega}_B \times \overrightarrow{\mathbf{V}}_B, \\ \overrightarrow{\mathbf{a}}_B^{\,n} &= \frac{\overrightarrow{\mathbf{V}}_B^{\,n+1} - \overrightarrow{\mathbf{V}}_B^{\,n-1}}{2\Delta t} + \overrightarrow{\Omega}_B^{\,n} \times \overrightarrow{\mathbf{v}}_B^{\,n}, \end{aligned} \qquad (7)$$

where $n+1$ and $n-1$ represent neighboring data points to point $n$, and $\Delta t$ is the time step (equivalent to $\frac{1}{240\,Hz}$). Note that for the correct approximation of the impact force, the effect of gravity has to be removed, such that

$a_i = |\vec{\mathbf{a}}_I - \vec{\mathbf{g}}|$ or $a_i = |\vec{\mathbf{a}}_B - \mathbf{R}_{BI} \vec{\mathbf{g}}|$, where the acceleration due to gravity is $\vec{\mathbf{g}} = [\,0 \quad 0 \quad 9.81\,]^\top$ m·s$^{-2}$.

To better comprehend the dynamics of reorientation and identify the dominant parameters, we examine the angular acceleration induced by the impact force's moment. The rotational motion of the robot can be represented as

$$\vec{\mathbf{M}}_B = \mathbf{I}_G \dot{\vec{\mathbf{\Omega}}}_B + \vec{\mathbf{\Omega}}_B \times (\mathbf{I}_G \vec{\mathbf{\Omega}}_B), \tag{8}$$

where $\mathbf{I}_G$ is the inertia tensor calculated about the body-fixed axes at the COG, and $\vec{\mathbf{M}}_B = \vec{\mathbf{r}} \times \vec{\mathbf{F}}$ is the net moment about the body-fixed axes, resulting from the external force $\vec{\mathbf{F}}$ applied at a moment arm $\vec{\mathbf{r}}$ from the COG. If we express the nose tip offset relative to the COG as $\vec{\mathbf{r}} = [\,l_{x_{off}} \quad 0 \quad -l_{z_{off}}\,]^\top$ (see Supplementary Fig. S2) and assume the robot is flying toward a vertical target with only a pitch angle of $\theta$ (zero roll and yaw angles, and no angular speeds), then the moment equation due to the impact force $F_i$ simplifies into the following pitching acceleration dynamics

$$\dot{q} = \ddot{\theta} = \frac{F_i \left( l_{z_{off}} \cos\theta + l_{x_{off}} \sin\theta \right)}{I_y}, \tag{9}$$

where $I_y$ is the robot's mass moment of inertia about the lateral axis.

### Static perching model

**Model assumptions.** The wing-wrapping model developed is based on some fundamental assumptions, with the primary one being that the statically perched UAV can be analyzed in a 2D plane rather than in the whole 3D space. The idea is to first analyze all the interactions between the robot and the pole on which it is perched and then move to the 3D space to calculate the maximum weight the system can hold.

Another critical point is that all the different segments are assumed to be touching the pole. This is not always the case as sometimes, one of the segments loses contact with the pole since the configuration with all the segments touching it is not an equilibrium position. Nevertheless, this assumption is valid if no extreme combinations of pole sizes, UAV wingspan, and spring moment are considered.

The torsion springs are modeled to exert loads linearly varying with angle, even if they operate slightly outside manufacturer values for linear operation. The model also assumes a Coulomb friction scheme at the contact points on a macro level and takes no consideration for wet or dry adhesion effects on a micro-scale. The total friction force is split unevenly in the horizontal and vertical directions, with its magnitude directly proportional to the applied normal load to the surface and independent of the contact area.

The model does not account for the wing hooks, a simplification driven by the unpredictable nature of the interactions resulting from the random engagement of hooks with pole surfaces or tree barks. Furthermore, it only supports symmetrical wing designs, but with a possibility of using uneven segmentation per wing. Even though the robot used in the study has two moving segments of equal length, the number of segments on each wing can be set to any number, and the length of each segment can be selected independently. The model takes the pole and UAV data as inputs, calculates the glider's geometrical configuration, computes all the forces, and determines iteratively whether remaining statically perched on the pole is possible. The data regarding the pole that the model needs are the diameter and the coefficient of static friction only. On the other hand, the data concerning the UAV include all the geometrical and physical parameters as well as the spring information.

**Friction model.** Before presenting the step-by-step iterative force estimation process, it is helpful to make a minor remark. The friction force at each contact point with the pole surface acts in two different directions. One component is tangential ($F_t$) and lies in the $x-y$ plane, and the other is vertical ($F_v$), parallel to the axis of the pole along $z$ axis (see Fig. 4a). The

former keeps the wings wrapped in place, avoiding slipping and detachment from the pole, while the latter is the force that overcomes gravity and prevents the robot from falling. The Coulomb friction model[45] gives

$$F_f = \mu F_n. \tag{10}$$

The combined friction coefficient, denoted as $\mu$, is a scalar value that represents the ratio of the total friction force $F_f$ to the force normal to the pole ($F_n$). Using vectorial sum and by denoting the tangential and vertical friction coefficients as $\mu_t$ and $\mu_v$, respectively, we can decompose Eq. (10) into

$$\begin{cases} \vec{\mathbf{F}}_f = \vec{\mathbf{F}}_t + \vec{\mathbf{F}}_v, \\ F_t = \mu_t F_n, \\ F_v = \mu_v F_n. \end{cases} \tag{11}$$

Given that the two components are perpendicular to each other, the coefficients must satisfy the condition

$$\mu^2 = \mu_t^2 + \mu_v^2. \tag{12}$$

At maximum payload capacity, the combined friction coefficient reaches its peak value of $\mu = \mu_s$, which is the critical threshold for the robot's ability to statically remain perched before the onset of slippage or falling.

**Solving scheme.** We first solve a purely geometric problem to find the contact points. The model positions a pole of diameter Ø with its center at the origin of the $x-y$ plane (Fig. 4a). Then, it assumes that the UAV and the pole will be tangent at each point of contact. The fuselage is always positioned tangent to the lowest point in the circumference. Then all the segments are added, starting from the innermost ones closest to the fuselage, leveraging the tangent constraint and the fact that they share a hinge with the fuselage or the previous segment. Once the relative positions of the pole and the segments are determined, we know the angles at each hinge and can compute the moments provided by the torsion springs using

$$M_s = k_s \theta_h, \tag{13}$$

where $k_s$ and $\theta_h$ are the stiffness of the torsion spring and hinge angle, respectively. The model solves the planar multibody problem starting from the outermost segment (denoted by number 2 in Fig. 4) since it is the only one affected by a single spring. Therefore, looking at this segment only, the normal force $F_{n_2}$ exerted from the pole to the UAV is calculated using the rotational equilibrium with respect to the hinge. Once the normal force is known, the in-plane component of the friction force $F_{t_2}$ is computed using the corresponding tangential friction coefficient $\mu_t$ as such

$$F_{n_2} = \frac{M_{s_2}}{l_{2,2}}, \tag{14}$$

$$F_{t_2} = \mu_t F_{n_2}, \tag{15}$$

where $l_{2,2}$ is the corresponding length of segment 2 from the hinge location $h_2$ to its contact point $c_2$. The model then advances to the next segment, solved similarly, always leveraging the moment equilibrium. The considered subsystem can be solved at each instance since the forces acting on the previous segments are already calculated earlier. For the sake of brevity, we only present the moment equilibrium equation in vector form, from which $F_{n_1}$ and $F_{t_1}$ can be computed

$$\vec{\mathbf{M}}_{s_1} + \vec{\mathbf{l}}_{1,1} \times \vec{\mathbf{F}}_{n_1} + \vec{\mathbf{l}}_{1,2} \times \vec{\mathbf{F}}_{n_2} + \vec{\mathbf{l}}_{1,2} \times \vec{\mathbf{F}}_{t_2} = 0, \tag{16}$$

$$F_{t_1} = \mu_t F_{n_1}. \tag{17}$$

This completes the calculation of all the normal and in-plane friction forces of a single wing, tailored to the specific configuration of the two-segment wing in this study. However, this successive segment analysis method is versatile and easily scalable to systems with more segments. By leveraging the symmetrical wing constraint, forces on the other half of the system can be easily resolved, leading to the final force equilibrium on the fuselage, which determines the last unknown normal force ($F_{n_0}$). This is achieved through force equilibrium along the $y$ direction (see Fig. 4a)

$$\overrightarrow{F}_{n_0} + 2\,\overrightarrow{F}_{n_1,y} + 2\,\overrightarrow{F}_{t_1,y} + 2\,\overrightarrow{F}_{n_2,y} + 2\,\overrightarrow{F}_{t_2,y} = 0 . \tag{18}$$

Lastly, we examine the solution obtained against specific constraints to ensure it is physically attainable. The solution is rejected if any of the normal forces exerted on the robot is negative, i.e., directed inwards towards the center of the pole, which is physically impossible. Additionally, an out-of-plane analysis compares the sum of vertical forces ($F_v$) against the weight ($W$). To ensure the robot can sustain its weight without falling, the following inequality must be satisfied

$$\mu_v\left(F_{n_0} + 2F_{n_1} + 2F_{n_2}\right) \geq W . \tag{19}$$

**Algorithmic process and friction split.** The only critical element remaining in the modeling process is understanding how the friction splits between the planar and vertical components. This cannot be explicitly estimated a priori based on the robot and pole data. To solve this issue, the model systematically sweeps the whole range of possible splits between the tangential and vertical friction coefficients and identifies all combinations that satisfy the force balance conditions for remaining perched. These conditions are namely having sufficient in-plane friction $F_t$ to prevent the wings from slipping off the pole (Eq. (18)) and having sufficient out-of-plane $F_v$ force to overcome the weight (Eq. (19)). Among potential solutions, the model reviews and selects the one with the minimum combined friction coefficient ($\mu$), corresponding to the most efficient use of friction force for the given conditions.

The model's flowchart, depicted in Supplementary Fig. S3, elucidates the friction split determination process. The process consists of an outer and an inner loop. The outer loop adjusts the split percentages between the tangential and vertical friction components ($\mu_t$ and $\mu_v$) while iterating through combined friction coefficient values ($\mu$) in the inner loop for a viable solution. For an input UAV weight with payload ($W$), the process starts by setting the fraction of the tangential friction ($\mu_t$, %) to zero percent. This fraction is incrementally increased up to 100% through successive iterations of the outer loop. Inside each iteration, the inner loop begins at zero combined friction coefficient percentage ($\mu_\%$) and ascends to 100%, corresponding to the theoretical limit of static friction coefficient ($\mu_s$). The loop proceeds by calculating $\mu_t$ and $\mu_v$ for each case and solving the multibody force and moment equilibrium problem (Eqs. (14) to (19)) to confirm if the friction split leads to a stable perch. If a solution is identified, the iteration ceases; otherwise, the loop continues with an incremented $\mu_\%$. Upon completion, the model selects the solution minimizing the total friction coefficient ($\mu$). This selection represents the optimal use of friction while maintaining a low error between the estimated vertical force and actual weight. Due to the iterative nature of the process, although the inequality in Eq. (19) confirms perching stability, it does not guarantee an accurate approximation of the vertical force. At times, it might overestimate the net vertical force compared to the weight, while true static equilibrium mandates an exact match between the two forces.

**Wing segmentation sizing**
The static model can aid with proper dimensioning of the wings at the design stage. While in theory it is the wingspan that sets the range of poles on which the robot can potentially perch, segmentation of the wing affects the static load-carrying capacity. The model serves as a valuable tool during the design stage in selecting the right segment size for a given number of segments. For PercHug, we selected two equal folding segments per wing

and explored three configurations by varying the fuselage and moving segment widths to achieve a constrained wingspan of 960 mm.

These considered wing configurations correspond to folding segments of width 205 mm, 195 mm, and 185 mm. Based on simulation results, we selected the second configuration which outperformed the others, exhibiting higher static payload capacity on most poles. This finding is illustrated in Supplementary Fig. S4c, where it is evident that the third configuration had limited success on poles, while the first and second configurations performed well, with the second one being the preferred choice. The simulation results are supported by experimental verification with the selected wing configuration (see Static perching experiments for more detail).

**Static perching experiments**
We validated the predicted wing design (see Wing segmentation sizing for configuration selection) with experiments of a physical prototype weighing 325 g attached to poles of different sizes and materials. The testing strategy involved placing the prototype robot, equipped with hook-less perching wings, on various poles and trees within the range specified by the model. We then gradually added calibration weights of 100 g each at the COG until the point of sliding down. We recorded the maximum weight that the prototype could sustain before falling as the static payload capacity (Fig. 5c). The measurements were repeated five times for each pole and showed no variation. The surface texture and specifications of the poles are listed in Fig. 5a and b (see Friction coefficient measurement for the methods used to estimate the coefficient of static friction). The tested objects included two indoor poles with diameters 250 mm and 315 mm, covered in copy paper, rubber pad, and paper towel (designated as poles I, II, and VI-IX), a 260 mm bamboo tree guard (IV), smooth and rough concrete columns of 350 mm (III and V), and six trees (X-XV) with diameters ranging from 265 mm to 360 mm. Note that diameters smaller than 265 mm invalidate the model assumptions, as either one wing has to overlap the other or the robot needs to slightly tilt in the vertical direction while the wingtips are touching, effectively reducing the wrapped diameter. While such corner cases fall outside the design and modeling scope, we still considered a few pole sizes slightly below this threshold as results matched well. The model predictions become further invalid as we deviate further from the minimum diameter limit.

**Friction coefficient measurement**
One of the two essential pieces of information needed to run the static perching model is the value of the static friction coefficient between EPP and the surface of the pole. The method for measuring this coefficient changes based on the specific case considered. For poles I, II, and VI-IX, the friction coefficient measurement was straightforward, as the materials covering these indoor poles could be detached from them for measuring purposes. We employed two methods to estimate these static friction coefficients, based on Coulomb's model principles[45]. The first method involved placing the material of interest on a flat surface and positioning a block of EPP foam (with known weight, $mg$) on top. We then used a force-measuring device, a spring balance, to gradually increase the pulling force on the EPP foam block, exerted parallel to the surface, until reaching the onset of movement ($F_{pull}$). The value of the static friction coefficient was then given by

$$\mu_s = \frac{F_{pull}}{mg} . \tag{20}$$

The second method was based on the friction angle $\theta$[45] concept, i.e., the maximum inclination angle that still prevents sliding. In this approach, the pole material is positioned on a flat horizontal plane with the foam block placed on top it. The surface is gradually inclined until the the block begins sliding. The static friction coefficient can be estimated by measuring this inclination angle using

$$\mu_s = \tan\theta . \tag{21}$$

We used both techniques for all the cases with the possibility of removing the surface material from the pole. We measured the static friction coefficient 10 times with each method. The final value was computed as the average of all measurements.

For the rest of the poles (mainly trees) in which the surface material could not be detached from the pole itself, the measurements had to be taken on a vertical surface. Since no standard testing procedure was found in the literature, we proposed a technique to do so. The concept involves pushing a piece of the material of interest with known weight ($mg$) against the vertical surface of the pole with a known force and pulling it upward ($F_{pull}$) using a force-measuring device until sliding. An essential requirement for such a technique is the ability to create a known normal force using elements that do not affect the object under measurement with their masses or other forces that could change the balance between them.

We designed the measuring tool in Supplementary Fig. S5 to tackle such an issue. It is composed of an aluminum frame with adjustable arms that tightly grip the pole and a foam block that is pressed against the surface of the pole using a set of four linear springs. The linear springs are placed inside small channels to keep them straight along the horizontal direction. Since the springs are the only connection between the block and the aluminum structure, the weight of the support structure is not passed to the block. Therefore, the measurement of the friction is not affected. The compression of the springs can be calculated by measuring the distance between the foam block and the fixed plastic plate, relative to their original lengths. Consequently, knowing the elastic pressing force, the static friction coefficient is computed by

$$\mu_s = \frac{F_{pull} - mg}{k\Delta l}. \tag{22}$$

where $k$ is the spring stiffness, and $\Delta l$ is the compression amount. To minimize errors with this measuring tool, we conducted 10 measurements on each pole or tree with non-removable material, by systematically repositioning the tool to account for surface or tree bark irregularities. The table in Fig. 5b presents the mean and standard deviation for the measured static coefficient of friction across all the studied poles.

**Dynamic perching experiments**

With the trigger strategy set to release upon primary impact, we used the PercHug prototype to conduct a minimum of five perching experiments on each tree for the two selected nose configurations: the standard upturned nose and the elastic extension with $D = 0.233 \, \text{N} \cdot \text{m}^2$. This resulted in a total of 80 perching trials, approximately 40 tests for each nose type across the six trees. We used a camera to record the perching events in slow-motion at 240 Hz. The speed and pitch angle data were extracted from the recorded footage using the Physlets Tracker software, by tracking the colored markers placed on PercHug.

For the perching characterization, we only considered experiments that met specific criteria. One crucial requirement was the successful execution of a reorientation maneuver, as unsuccessful reorientation would inevitably lead to unsuccessful perching. Previous findings of the Inertial reorientation section indicated that a minimum impact angle of 15° and 8° was necessary to ensure successful reorientation with standard and elastic noses, respectively (Fig. 3c). Therefore, trials that did not meet these conditions were excluded. Additionally, tests in which the impact occurred on the fuselage or wings instead of the nose, resulting in no or unsuccessful reorientation, were also excluded. As a result, 16 out of 80 trials were disregarded; for a detailed breakdown, refer to Supplementary Table S1.

**Data availability**

All data needed to evaluate the conclusions of the paper are available in the main manuscript and the supplementary information.

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

## Acknowledgements

This work was supported in part by NCCR Robotics, a National Centre of Competence in Research, funded by the Swiss National Science Foundation (Grant no: 51NF40_185543), and by the European Union's Horizon 2020 research and innovation program through the AERIAL-CORE project (grant no: 871479).

## Author contributions

M.A., W.S., A.J.I., and D.F. conceptualized the ideas and organized the study. M.A. and M.B. designed and fabricated the robot prototypes. M.A. conducted the reorientation experiments and analyzed the data (with input from H.V.P. and W.S.). M.B. developed the static model and performed static perching experiments (with input from M.A. and W.S.). M.A. and M.B. conducted the dynamic perching experiments with the PercHug robot and analyzed the data. M.A., M.B., H.V.P., W.S., A.J.I, and D.F. wrote and revised the manuscripts.

## Competing interests

The authors declare no competing interests.
