## [Peer Review File · Communications Engineering]

Reviewers' comments:

Reviewer #1 (Remarks to the Author):

- 1) In fact, it can be observed that the legs of some animals also play a considerable role as they perch on trees in a wrapped posture, such as bats and owls in Figure 1. How do authors think about it.
- 2) The limitation of this aircraft is that it can only complete one-way flight to perching, and can not be detached from the tree and continue to fly, which is a one-time use. How do the authors think?
- 3) There are no electronic components, control modules and other driving parts in the whole robot, and the maneuvering flight before perching is completed only by manual or other external devices, which limits the movement field. Therefore, it is suggested that the avionics of the robot should be improved to enable the robot to perch on the tree after autonomous flight.
- 4) At present, the robot can only realize the hand-launched to the tree to perch, and can not move or return from the tree, so there is still a certain distance from the future outlook in the Discussion.

Other detailed comments are as below:

- 1) Page 2, line 116: "Two torsion springs with a stiffness of ..." While there appear to be 3 torsion springs in the PerchHug robot in Figure 1.
- 2) Page 2, line 119: "Upon impact, the springs are released..." Which springs are released? The springs in Figure 2d or g? It's confusing.
- 3) Page 2, line 130: "a dihedral angle of about 5° in order to improve the lateral stability of the aircraft." Why a dihedral angle of 5° can guarantee the lateral stability of the aircraft?
- 4) Page 3, line 152-155: How does the released latch move? If it would affect the stability of the vehicle.
- 5) Page 4, Figure 3: The standard and elastic nose are best depicted in Figure 3 respectively, which is easier to identify.
- 6) Page 4, line 159-160: The elastic nose in Supplementary Video is sometimes disappeared. It is suggested to demonstrate the dynamic perching performance on trees of two kinds of noses on four kinds of tree trunks (X-XV).
- 7) Page 7, Figure 6: Figure 6d and e represent the same data and it is recommended to combine them together.
- 8) Page 7, line 306, Page 9, line 552: There are 550g and 220g for the weight of the robot prototype? It is best to list the weight of the parts of the robot prototype.
- 9) Page 7, line 315-316: In all dynamic perching experiments, authors hand-launched the robot toward the trees, which is not controllable for the impact speed of each test, and it cannot be ensured that each test is consistent.
- 10) Page 8, line 380-384: These data cannot be found out from the graph.
- 11) Page 8, line 452: How does the bistable mechanism with the trigger point on the ventral side of the robot respond to the first impact and release the wing to wrap pole?
- 12) Page 9, line 542: Please ensure the unit of "0.2 mm, 20 kg" is correct.
- 13) Page 11, line 725: Why the lowest net friction coefficient is the final solution?
- 14) Page 11, line 730, line 739: The net tangential friction coefficient is still expressed by t subscript instead of h .
- 15) Page 11, line 746-750: Please explain the concepts such as total friction coefficient, net friction coefficient of, and illustrate that why the minimum or maximum it is to be achieved.

- 16) Page 12, Equation (16) and (17): These two methods are commonly used to measure the friction coefficient, and can be quoted from authoritative books.
- 17) Page 12, Equation (18) and Figure S5: This self-designed measuring tool will have a certain error, how to judge it?
- 18) Page 12, line 863-864: How to build the motion capture system outdoor?
- 19) The title of Figure S5 should be “b” rather than “c”.
- 20) Supplementary Video should be a dynamic video that is not convenient to be expressed in the text, but the authors put some pictures in the text into the Supplementary Video, which is unreasonable. It is suggested that the authors could refer to the content of the supplementary video in some high-level papers.

Reviewer #2 (Remarks to the Author):

This presents a bioinspired winged robot capable of robust crash-landing and perching on vertical poles and trees. The design uses an upturned nose and hinged, folding wings to enable passive reorientation and wrapping upon impact without complex control. You systematically characterize the inertial reorientation, static perching capacity, and dynamic perching success through modeling and experiments. The capabilities could enable new applications in inspection, infrastructure maintenance, and environmental monitoring.

Overall, I find this an interesting study presenting a useful advancement in morphological adaptation and aerial manipulation. The analysis provides valuable insights for vehicle designs. However, some issues should be addressed:

- I believe in the entire manuscript, the authors have not adequately investigated the contribution of the hooks. Early on, the manuscript indicates the presence of hooks at the wing tips. However, in the subsequent modeling, the hooks were ignored (which is fine, but it should be made a little clearer). Then, in the sdynamic perching validation, I believe the hooks were present. This distinction should be clearly mentioned. Then, the authors claim that the results match the models. Will they still match the models if the hooks were absent? It would be informative and helpful if the usefulness of the hooks is systematically verified.

- Related to the comment above, the fact that the static perching experiments were carried out with hook-less perching wings should be clear in the main text (not just in the Methods).

- From my understanding, the force estimation (under Methods: Flight kinematics and impact force estimation) may not be technically correct. Let me elaborate and see if you disagree (I'm happy to change my view).

>> In short, what you claim to care about is the accleration of the robot in the body frame. This is because you finally use a_i in (7), that is $F = m \cdot a_i$. Hence, why do you not just numerically take the derivatives of the position twice?

>> Instead, you first calculated the velocity of the robot, in the body-fixed frame, by reprojecting \dot{P} . This becomes V in (2).

>> The derivative of V in (6) is the acceleration in the body-fixed frame. This is not "a" in (6). In fact, $F =$

$m(dV/dt + \Omega \times V)$. This is force in the body frame. However, I believe you want to obtain force in the inertial frame.

>> Furthermore, this section is not extremely well organized. For example, you never use (3) (you only use the rotation matrix and euler angles, not the angular velocity). The sentence starting with "Any rigid body, such as flying robot, ..." at the end of the first paragraph and the sentence "Multiplying any vector defined in ..." are not really needed (they are very fundamental).

>> The recalculated accelerations may not be that different from the previous values, but it is important to make sure that the paper is technically sound.

Less critical points (but may still be important):

- Page 2, Line 120: "This gripping force, combined with the friction along the vertical axis resulting from the gravitational force, ..." >> I'm confused. What do you mean when you state that the gravitational force is resulted from the friction?

- Page 5, Line 242: I totally understand why the authors state that the maximum angular limit is 180 deg (which may sounds intuitive), but is this really a theoretical limit? I imagine is the friction coefficient is very high, perching should be feasible at larger angles (at least in theory)?

- Fig. 5c, please define "sensitivity".

- Fig. 6d, it would be useful to include absolute numbers with the success rates.

- Page 7, Line 335, the number 20 ms delay looks more like 25 ms in Fig. 6b.

- Page 9, Line 542, Dyneema is a tradename. It might be useful to replace that with "composite fiber? (Dyneema)".

- Page 9, Line 520, "tough" >> "tough"

- Discussion, I suppose the ability to take off afterwards should be in the future direction?

Addressing these concerns, especially clearly conveying the usefulness of the wing hooks, will significantly enhance the manuscript. I look forward to seeing a revised version of this work.

Reviewer #3 (Remarks to the Author):

Summary:

The authors present a perching solution for small fliers in which a fixed-wing flier performs a controlled pitch-up maneuver and wraps its wings around a slim tree trunk or approximately vertically aligned branch in a hugging grasp. The maneuver just prior to landing is not dissimilar to that from Lussier-Desbiens et al. [2012] although the enveloping grasp is certainly more secure than hanging from feet with spines as in the previous work.

In summary, the paper presents an interesting alternative concept for perching on approximately vertically oriented tree trunks and branches. The references to previous related work and the dynamic modeling are appropriate and provide a good argument for future researchers to consider the authors' proposed approach. The overall writing style is clear and the illustrations are excellent, giving the reader a clear sense of what the wing-wrapping glider can do. The video and supplementary material are

welcome.

Nonetheless, the work is in an early stage of development with respect to a complete solution. To put the work in perspective, the authors should talk about some of the extensions and how they would be accomplished. For comparison, if this work were submitted to IEEE, it would likely be considered suitable for ICRA or IROS, and potentially for RA-Letters with some attention to the items below, but not for TRO or a similar journal.

Operational questions and concerns:

The plane in this paper is a glider. Is this proposed as the long term solution? Will gliders be released from a powered plane and then controlled to find perching sites? Alternatively, if the long-term solution is that wing wrapping will be used with a powered plane, where will the propellor go? Perhaps a rear propulsion scheme or with wing-mounted propellers? If the authors do not intend for the gliders to be used as the long term solution they should comment on this.

The paper claims a 71% success rate for perches; this is actually not particularly high. What happens to the approximately 30% that fail to perch? Do they fall to the ground? If the plane were powered there could be an emergency recovery solution (as in some previous work on perching fliers) – but that seems less likely with a glider.

The paper covers relationships among nose geometry and stiffness and perching maximum forces and success rate for a variety of approach velocities, pitch angles, tree trunk diameters and materials. There is no discussion about the effects of errors in targeting in the lateral direction (i.e., making contact somewhat to the left or right of the centerline of the tree trunk) and no discussion about angular errors in the yaw and roll degrees of freedom. Controlled experiments for this prototype were evidently conducted indoors with motion capture, where such errors may be kept small. But the intended use is outdoors, where air currents will make such errors unavoidable. One such alignment error appears to occur at about 2:09 in the attached video.

Thus in the longer term it will be necessary to establish the “envelope” of initial conditions in targeting, as well as velocities, for all degrees of freedom.

Detailed comments on the manuscript:

I find figure 3c confusing. Looking at the data and variations in the data I cannot discern a clear correlation among design parameters, success rate and maximum impact forces apart from the fact that success rate increases and variability decreases with beta. The claimed optimality of $D = 0.233 \text{Nm}^2$ is not strongly supported. The authors write: “The improved success rate may be a result of a longer moment arm from the impact point to the COG of the robot”, which is speculative. But this should be fairly straightforward to test either based on a kinematic simulation or empirically.

A further discussion about the considerations for autonomous flight would make the paper stand out more. Beyond discussing that this is future work, some consideration for what sensors/planning/ etc method would enable autonomous perching would be useful to the reader.

The wing cannot be automatically removed from the tree branch. What are some of the challenges in doing so? This also ties in with the need for a recovery strategy including how to detect failure and how to recover from it without falling to the ground.

A low impact angle leads to lower success rate. How could the design be improved to improve perching at low impact angle?

Minor points

Fig. 5c refers to error bars as 'sensitivity' - a better word would be variability (perhaps 1 standard deviation?).

Figure 4 and related discussion: What exactly are the min and maximum wrap angles? It looks like the limits are from 180 degrees up to some number less than 360 degrees? What determines the maximum wrap angle? Further, in Fig. 5c - why does the solution sometimes work for diameters that are smaller than the theoretical smallest diameter (250 mm vs. 265mm)?

The static modeling seems mainly to inform us that the drone will not fall once it has successfully wrapped around a tree. It seems that the majority of failures will arise from dynamic interaction between the drone and the tree and an analysis of this dynamic interaction would be more informative. It may lead to insights regarding trajectories, spring stiffnesses for the torsional springs and triggering sensitivity that will increase the probability of success despite variations in targeting and velocity, as will occur in the field.

Crash-Perching on Vertical Poles with a Hugging-Wing Robot

Mohammad Askari, Michele Benciolini, Hoang-Vu Phan, William Stewart,

Auke J. Ijspeert and Dario Floreano

The authors thank the reviewers for their careful reading of the manuscript. The comments were extremely helpful in revising the paper and making the contribution more solid. Responses to each comment are given with the original comment. For convenience, where changes were made to the manuscript, the changes are shown in red text and reproduced here with the response.

Reviewer 1 Comments

- In fact, it can be observed that the legs of some animals also play a considerable role as they perch on trees in a wrapped posture, such as bats and owls in Figure 1. How do authors think about it.

We appreciate the reviewer's comment regarding the role of hind-limbs and posture. It is an insightful perspective that highlights the versatility of strategies employed by animals for holding onto different surfaces. Although many animals use legs to support perching or resting on trees, we avoid using additional leg structures to mitigate mechanical complexity and increased mass. Nevertheless, we have carefully surveyed various studies for the roles of feet, toe arrangements, and tails in perching animals, and have added references for the interested readers. The revised text also reflects our design choices based on some common principles found in nature; an example of which is how a long tail helps during the crash-landing maneuver, as seen in geckos [28], or helping against pitch-back motion while resting on trees as seen in birds [30,31], due to the longer moment arm to the gripping force generated by the wings.

The UAV then leverages foldable, pre-loaded segmented wings, which are released through a latch system at impact, to wrap around vertical poles for perching. This behavior imitates that observed in certain flying animals (see Fig. 1). While studies have examined morphological adaptations and use of wings, feet, and tails of select bat [29] and bird [30-32] species, a broader observation reveals common principles among all perching and climbing animals [33]. Our solution avoids dedicated perching feet structures that increase body mass and complexity, opting instead for a dual-use strategy leveraging existing UAV elements. This includes employing our front limbs (wings) to tightly hug poles and maintain the center of mass close to the pole to minimize the pitch-back effect (Fig. 1). The use of a long tail is also found to be effective in nature for both landing [28] and resting [30,31]. Moving forward, we provide the design details of the upturned nose and wing elements, investigate the performance of inertial re-orientation and wing wrapping induced by collisions, and validate the crash-perching capability on tree trunks using PercHug, a gliding-winged robot.

- The limitation of this aircraft is that it can only complete one-way flight to perching, and can not be detached from the tree and continue to fly, which is a one-time use. How do the authors think?

We acknowledge the current limitation of our aircraft and agree that an unperching solution and a fly-away maneuver are valuable aspects to explore further. We are currently working toward these tasks and plan to report our results in an upcoming work in the future. As in this paper, we mainly focus on a novel and effective perching method for winged robots. In response to this valuable feedback, we have extensively revised the discussion section of the manuscript to explicitly address the limitations of our platform and outline our vision for potential solutions. We are actively researching the development of a powered system that incorporates sophisticated controlled maneuvers to achieve unperching and subsequent flight. Screenshots of relevant parts of this discussion are included here, and we invite the reviewer to refer to the updated manuscript for the complete discussion. We thank the reviewer for the input, and these refinements aim to provide a more thorough exploration of the challenges and future directions associated with our robotic platform.	While we successfully demonstrated a fully passive pole perching method, our platform currently lacks flight and targeting capabilities prior to perching, along with the ability to unperch and take flight again. To address ver. Given the preloaded and hugging design of the wings, a crucial enhancement to the system involves transforming the latch-release mechanism into a revertible powered subsystem, enabling active control over wing opening and closing. This modification unlocks a range of new functionalities. Firstly, it can facilitate thrust-assisted climbing by slightly opening the wings to loosen the gripping force, allowing movement across the pole. A quick release capability enables a seamless return to perching mode upon reaching the target spot on the pole. Secondly, it facilitates full wing opening for unperching, while thirdly, aiding recovery from a failed perching event identified by the IMU to prevent a fall to the ground. Executing a controlled recovery maneuver involves reopening the wings quickly, coupled with using thrust and adequate control authority from aerodynamic surfaces to regain stable flight. Overcoming the high spring forces at the wing joints presents a challenge in designing this active wing opening mechanism, which necessitates using a slow high-torque motor to pull onto the tensioning wires given their short moment arm with the springs. Extending this arm and guiding the wire across higher points on the wings can alleviate this issue. Additionally, the slower wing opening rate compared to wing closing requires a solution for remaining attached to the pole during unperching. Potential strategies involve utilizing thrust for momentary hovering, incorporating retractable hooks under the fuselage for engagement, or utilizing materials with bidirectional friction properties to facilitate upward movement while preventing downward falls. These advancements will mark a pivotal step in overcoming existing limitations, contributing substantially to the evolution of a more comprehensive aerial system.
---	---

- There are no electronic components, control modules and other driving parts in the whole robot, and the maneuvering flight before perching is completed only by manual or other external devices, which limits the movement field. Therefore, it is suggested that the avionics of the robot should be improved to enable the robot to perch on the tree after autonomous flight.

We appreciate the reviewer's feedback regarding the absence of electronic components, control modules, and autonomous flight capabilities in our robot. It is important to highlight that integrating avionics for autonomous flight and pole detection involves substantial additional work. However, we have carefully addressed the lack of discussion on this matter in the revised text. Our primary objective in this study was to explore the crash-perching maneuver while understanding its mechanics. The results here provide crucial insights into the flight conditions necessary for successful perching (such as approach angle, speed, and target poles), which serve as foundational knowledge required for the development of control schemes for autonomous flight and perching in our follow-up research.	While we successfully demonstrated a fully passive pole perching method, our platform currently lacks flight and targeting capabilities prior to perching, along with the ability to unperch and take flight again. To address these shortcomings, future developments based on this work will integrate avionics and proper control surfaces to enable autonomous flight and perching. Ongoing efforts involve sensor-based pole detection for perching, a powered grip loosening control for thrust-assisted climbing, and unperching, collectively contributing to a complete mission cycle. In the planning phase of a perching mission, a GNSS-receiver can provide means to autonomously navigate from far away to a target region within a few meters of accuracy [38]. Subsequently, a target pole can be detected using vision-based methods from tens or hundreds of meters away [39–41]. Switching to an accurate vision-based flight control method can then facilitate maneuvering to the desired spot, as demonstrated in [42]. The success or failure of the perching maneuver can be determined using data from an inertial measurement unit (IMU). Additionally, the propulsion scheme is a critical consideration, favoring a rear fuselage-mounted one due to impracticality in mounting propellers on the nose or wings. The incorporation of a propulsion system also holds potential for further enhancing perching success by utilizing the thrust to push toward the pole and against gravity to reduce the bounce-off effect during the landing maneuver.
---	---

- At present, the robot can only realize the hand-launched to the tree to perch, and can not move or return from the tree, so there is still a certain distance from the future outlook in the Discussion.

This is quite right. And as with the reviewer’s previous comment on avionics, reaching the future outlook that we describe will require considerable effort beyond this paper. However, in order to get to that point, a complete understanding of the mechanics of the impact, reorientation, and wing-wrapping is critical. That is the insight provided by this paper. We include such a far-reaching future outlook because we believe that this paper will have a far-reaching impact.

Detailed Comments

- Page 2, line 116: “Two torsion springs with a stiffness of ...” While there appear to be 3 torsion springs in the PerCHug robot in Figure 1.
- Page 2, line 119: “Upon impact, the springs are released...” Which springs are released? The springs in Figure 2d or g? It's confusing.

Yes, the reviewer is indeed correct about the number of springs being three. We are thankful for the correction and have corrected this in the text and the CAD view (Fig. 2d). We have also made clarification about the combined spring stiffness and that we mean the torsion springs on the wings (Fig. 2d) are released for wing-wrapping.

The vehicle is equipped with foldable wings that have three hinged segments. One segment is attached to the fuselage, while the other two can bend in the ventral direction to wrap around the pole (Fig. 2d). Three torsion springs, in parallel configuration with a combined stiffness of $3.45 \text{ N}\cdot\text{mm}/^\circ$, are placed at the interface between the two segments and are pre-loaded during flight. Upon impact, these springs are released and cause the segments to fold and press against the pole. This gripping force, com-

- Page 2, line 130: “a dihedral angle of about 5° in order to improve the lateral stability of the aircraft.” Why a dihedral angle of 5° can guarantee the lateral stability of the aircraft?

Indeed 5° on its own will not guarantee stability, however, the use of a few degrees of dihedral is incredibly common on manned and unmanned aircraft across a broad range of scales. For this paper, 5° was picked somewhat arbitrarily based on typical dihedral angles, and stability is not guaranteed, but was sufficient for the experiments conducted. That is why we state an “improvement” in lateral stability and not suggest an “absolute guarantee”.

- Page 3, line 152-155: How does the released latch move? If it would affect the stability of the vehicle.

The latch (red piece in Fig. 2f) is tiny and weighs less than a gram. When released, it does not impact stability because of its low weight and the fact that it remains connected to the wires from the wings, which let it move just a little within the space inside the fuselage.

- Page 4, Figure 3: The standard and elastic nose are best depicted in Figure 3 respectively, which is easier to identify.

Certainly. The in-text reference has been corrected by changing it from (Fig. 2) to (Fig. 3).

- Page 4, line 159-160: The elastic nose in Supplementary Video is sometimes disappeared. It is suggested to demonstrate the dynamic perching performance on trees of two kinds of noses on four kinds of tree trunks (X-XV).

We have created additional supplementary videos, one of which (S1) specifically showcases dynamic perching experiments comparing the two types of noses on the same trees. The video is also appropriately referenced in the revised manuscript and the supplementary PDF where a relevant discussion is made.

perching tests. In these experiments, we tested PercHug equipped with and without the extended elastic nose of $D = 0.233 \text{ N}\cdot\text{m}^2$ (corresponding to the best reorientation performance) on the six trees used in the static perching experiments (trees X-XV in Fig. 5).

In all dynamic perching experiments, we hand-launched the robot toward the trees. A perching trial was considered successful if it involved four distinct phases of gliding, reorienting at impact, wrapping the wings, and staying perched on the tree (see Fig. 6a and Supplementary Video S1). We first investigated the effect of Video S1. **Dynamic perching – different nose types. This video showcases successful perching experiments with PercHug on various trees, using the standard upturned and extended elastic nose types.**

- Page 7, Figure 6: Figure 6d and e represent the same data and it is recommended to combine them together.

We have moved the table (Fig. 6d) to the supplementary material, expanded it with additional statistical data, and retained Fig. 6e (now labeled as Fig. 6d) in the main text, which now includes static friction coefficients for completeness.	Perching was more successful with the standard up-turned nose compared to the elastic nose on the majority of tested trees (Fig. 6d). The standard nose configuration yielded an overall perching success rate of 73% across all trials, significantly outperforming the 42% achieved with the elastic nose (see Supplementary Table S1 for a detailed breakdown). While the elastic nose improved reorienta-     tree properties standard nose (%) elastic nose (%)     XIII Ø: 265 μ: 0.98 100 25   XV Ø: 270 μ: 1.04 60 15   XII Ø: 280 μ: 0.85 100 85   XI Ø: 320 μ: 0.83 75 25   XIV Ø: 350 μ: 1.02 65 65   X Ø: 360 μ: 0.77 50 40   	tree properties	standard nose (%)	elastic nose (%)	XIII Ø: 265 μ: 0.98	100	25	XV Ø: 270 μ: 1.04	60	15	XII Ø: 280 μ: 0.85	100	85	XI Ø: 320 μ: 0.83	75	25	XIV Ø: 350 μ: 1.02	65	65	X Ø: 360 μ: 0.77	50	40
tree properties	standard nose (%)	elastic nose (%)																				
XIII Ø: 265 μ: 0.98	100	25																				
XV Ø: 270 μ: 1.04	60	15																				
XII Ø: 280 μ: 0.85	100	85																				
XI Ø: 320 μ: 0.83	75	25																				
XIV Ø: 350 μ: 1.02	65	65																				
X Ø: 360 μ: 0.77	50	40																				

- Page 7, line 306, Page 9, line 552: There are 550g and 220g for the weight of the robot prototype? It is best to list the weight of the parts of the robot prototype.

We have updated Fig. 3c to now include a comprehensive weight breakdown of the principal elements constituting PercHug. Additionally, we have clarified in the revised Methods section that the prototype used for the reorientation experiments was solely built out of foam and had identical dimensions to the main robot, but without any frame reinforcement or hardware components besides the nose types being tested.	hand-launched against trees. PercHug weighed 550 g (see Fig. 2c for the weight distribution), including the unlatching mechanism, bistable backup trigger, and folding wings with hooks (Fig. 2d-g), as well as a reinforced tail and body to enhance durability during multiple crash-perching tests. In these experiments, we tested PercHug weight: 550 g  ■ wings ■ tail ■ frame ■ bistable trigger ■ latch release Architecture of the PercHug platform. a Operating primary impact, (3) reorientation and wing release, (4) seconds of the impact forces, proportionally drawn. b Isometric physical properties of the robot. d Pre-loaded segmented We used a 220 g fixed-wing glider, made entirely out of EPP foam with dimensions given in Fig. 2c, and equipped it solely with these nose types to eliminate potential effects due to other UAV elements. We launched the robot with a
---	--

- Page 7, line 315-316: In all dynamic perching experiments, authors hand-launched the robot toward the trees, which is not controllable for the impact speed of each test, and it cannot be ensured that each test is consistent.

Hand-launching indeed introduces variability in impact speed, which is expected when conducting experiments outdoors. Our earlier indoor experiments were designed for systematic characterization and repeatability, while the outdoor tests served	PercHug successfully demonstrated crash-perching capability on all trees for impact speeds V_i ranging from 3 m/s to 5 m/s and relative impact angles β above 15°, regardless of the nose type (Fig. 6c). The success maps illustrate the distribution of successful and failed perching trials across different trees. These experiments occurred to have very similar average impact speeds of 4.1 ± 0.7 m/s for both nose configurations, indicating a notable consistency across the conducted trials despite hand-launching.
--	---

as rigorous validations, aiming to strengthen the perching assessment under real-world conditions. Despite the variability challenge, we aimed for consistent impact speeds across all trials (3 to 5 m/s). Remarkably, experiments with the two noses exhibited similar average speeds, which reflects a commendable level of repeatability.

- Page 8, line 380-384: These data cannot be found out from the graph.

As mentioned, the table in Fig. 6d now appears as Supplementary Table S1. We have expanded it to include the timing metrics data (mean triggering and wrapping times), which were missing from the original graph/table. We have also corrected minor numerical errors in the values presented in the revised manuscript.

success rate. The average trigger delays from impact were 26 ± 7 ms and 37 ± 15 ms, for the standard and elastic noses, respectively. In successful trials, the respective mean wrapping times were 156 ± 30 ms and 152 ± 25 ms, with a variation of less than 3% for the standard and elastic noses (see Supplementary Table S1). The comparable wrapping times imply that this parameter is likely a characteristic of the wings and unaffected by other UAV components.

Table S1 Dynamic perching statistics with PercHug. The table shows perching outcomes on different trees, characterized by their diameters \varnothing and static friction coefficients μ_s . The presented data includes perching success rates, expressed as percentages and actual number of experiments conducted, with the standard upturned and the extended elastic nose configurations. Additionally, the timing metrics for triggering and wrapping events are provided, measured from the impact instance.

tree	\varnothing (mm)	μ_s	success rate		triggering (ms)		wrapping (ms)			
			standard	elastic	standard	elastic	standard	elastic		
X	360	0.77	50%	(3/6)	38%	(5/13)	26 ± 5	39 ± 19	175 ± 33	163 ± 28
XI	320	0.83	75%	(3/4)	25%	(1/4)	30 ± 6	42 ± 14	146 ± 18	150 ± 0
XII	280	0.85	100%	(4/4)	86%	(6/7)	24 ± 5	31 ± 12	119 ± 16	138 ± 13
XIII	265	0.98	100%	(4/4)	25%	(1/4)	23 ± 7	42 ± 10	164 ± 21	200 ± 0
XIV	350	1.02	67%	(2/3)	67%	(2/3)	26 ± 6	24 ± 6	150 ± 6	131 ± 9
XV	270	1.04	60%	(3/5)	14%	(1/7)	27 ± 10	40 ± 15	192 ± 19	171 ± 0
			73%	(19/26)	42%	(16/38)	26 ± 7	37 ± 15	156 ± 30	152 ± 25

- Page 8, line 452: How does the bistable mechanism with the trigger point on the ventral side of the robot respond to the first impact and release the wing to wrap pole?

The bistable mechanism remains in its stable bottom-side-extended position and does not react to the primary impact. It only engages with the secondary impact, as discussed on pages 3-4, when it switches position (moves upward) to release the latch and deploy the wings. By adapting the latch mechanism to release at primary impact, the bistable mechanism essentially becomes redundant, as the latch is released beforehand.

- Page 9, line 542: Please ensure the unit of “0.2 mm, 20 kg” is correct.

The values and units were correct, but we have clarified what we meant by each value. We thank the reviewer for their attention to details.

diameter of 5 mm. Lastly, two eye screws are fixed to 3D printed pieces on the wing tips through which we pass the Dyneema cord, with a thickness of 0.2 mm and a maximum load capacity of 20 kg, that connects to the latch in the fuselage.

- Page 11, line 725: Why the lowest net friction coefficient is the final solution?
- Page 11, line 730, line 739: The net tangential friction coefficient is still expressed by t subscript instead of h.

- Page 11, line 746-750: Please explain the concepts such as total friction coefficient, net friction coefficient of, and illustrate that why the minimum or maximum it is to be achieved.

We thank the reviewer for pointing out the ambiguities in the static model presentation. To address each concern comprehensively, we have carefully refined and expanded relevant subsections under Methods. Key changes are highlighted in the screenshots. For a comprehensive overview, we direct the reviewer to the revised manuscript.

Throughout the revised text, we have improved the terminology used for the friction coefficients. The terms 'total friction coefficient' and 'net friction coefficient,' which essentially conveyed the same concept, have been replaced with the more explicit term 'combined friction coefficient.' This term now consistently represents the combined (vectorial) sum of the tangential and vertical friction components, whose subscripts are consistently applied across the text, figures, and tables. We have also clarified the physical implications behind these coefficients, particularly highlighting their relationship to the static friction coefficient and maximum payload.

Furthermore, we have expanded the discussion and equations related to multibody equilibrium analysis as well as providing a more in-depth understanding of the friction split determination process. In response to your query, we emphasize that selecting the lowest combined friction coefficient as the solution ensures the robot is using only the necessary amount of friction force to prevent slippage or falling, avoiding cases with potential overestimation of it compared to the actual weight.

The combined friction coefficient, denoted as μ , is a scalar value that represents the ratio of the total friction force F_f to the force normal to the pole (F_n). Using vectorial sum and by denoting the tangential and vertical friction coefficients as μ_t and μ_v , respectively, we can decompose Eq. (8) into

$$\begin{cases} \vec{F}_f = \vec{F}_t + \vec{F}_v, \\ F_t = \mu_t F_n, \\ F_v = \mu_v F_n. \end{cases} \quad (9)$$

Given that the two components are perpendicular to each other, the coefficients must satisfy the condition

$$\mu^2 = \mu_t^2 + \mu_v^2. \quad (10)$$

At maximum payload capacity, the combined friction coefficient reaches its peak value of $\mu = \mu_s$, which is the critical threshold for the robot's ability to statically remain perched before the onset of slippage or falling.

Lastly, we examine the solution obtained against specific constraints to ensure it is physically attainable. The solution is rejected if any of the normal forces exerted on the robot is negative, i.e., directed inwards towards the center of the pole, which is physically impossible. Additionally, an out-of-plane analysis compares the sum of vertical forces (F_v) against the weight (W). To ensure the robot can sustain its weight without falling, the following inequality must be satisfied

$$\mu_v (F_{n_0} + 2F_{n_1} + 2F_{n_2}) \geq W. \quad (17)$$

Algorithmic process and friction split. The only critical element remaining in the modeling process is understanding how the friction splits between the planar and vertical components. This cannot be explicitly estimated a priori based on the robot and pole data. To solve this issue, the model systematically sweeps the whole range of possible splits between the tangential and vertical friction coefficients and identifies all combinations that satisfy the force balance conditions for remaining perched. These conditions are namely having sufficient in-plane friction F_t to prevent the wings from slipping off the pole (Eq. (16)) and having sufficient out-of-plane F_v force to overcome the weight (Eq. (17)). Among potential solutions, the model

The model's flowchart, depicted in Supplementary Fig. S3, elucidates the friction split determination process. The process consists of an outer and an inner loop. The outer loop adjusts the split percentages between the tangential and vertical friction components (μ_t and μ_v) while iterating through combined friction coefficient values (μ) in the inner loop for a viable solution. For an

	with an incremented $\mu\%$. Upon completion, the model selects the solution minimizing the total friction coefficient (μ). This selection represents the optimal use of friction while maintaining a low error between the estimated vertical force and actual weight. Due to the iterative nature of the process, although the inequality in Eq. (17) confirms perching stability, it does not guarantee an accurate approximation of the vertical force. At times, it might overestimate the net vertical force compared to the weight, while true static equilibrium mandates an exact match between the two forces.
--	---

- Page 12, Equation (16) and (17): These two methods are commonly used to measure the friction coefficient, and can be quoted from authoritative books.

We have addressed this concern by referencing an authoritative book chapter on Coulomb's friction model and associated concepts, such as the friction angle. Additionally, we have refined the text for enhanced clarity. The concise inclusion of these methods is intentional, as we aim to make the paper entirely self-contained, recognizing that not all readers may be familiar with these techniques. Furthermore, their presentation solely within the Methods section ensures that the flow and clarity of the manuscript remain intact.	forward, as the materials covering these indoor poles could be detached from them for measuring purposes. We employed two methods to estimate these static friction coefficients, based on Coulomb's model principles [44]. The first method involved placing the material of interest on a flat surface and positioning a block of EPP foam (with known weight, m) on top. We then used a force-measuring device, a spring balance, to gradually increase the pulling force on the EPP foam block, exerted parallel to the surface, until reaching the onset of movement (F_{pull}). The value of the static friction coefficient was then given by $\mu_s = \frac{F_{pull}}{mg}. \quad (18)$ The second method was based on the friction angle θ [44] concept, i.e., the maximum inclination angle that still prevents sliding. In this approach, the pole material is positioned on a flat horizontal plane with the foam block placed on top it. The surface is gradually inclined until the the block begins sliding. The static friction coefficient can be estimated by measuring this inclination angle using
--	---

- Page 12, Equation (18) and Figure S5: This self-designed measuring tool will have a certain error, how to judge it?

Certainly, the reviewer's observation is accurate. To account for potential errors, we had conducted 10 repeated measurements of the static friction coefficient on tree trunks, repositioning the tool each time to account for bark irregularities. Although the results were previously listed under the Fig. 5b table, a discussion on this matter was indeed missing from the original manuscript. It is now added to the Methods section.	$\mu_s = \frac{F_{pull} - mg}{k\Delta l}. \quad (20)$ where k is the spring stiffness, and Δl is the compression amount. To minimize the error with this measuring tool, we conducted 10 measurements on each pole with non-removable material, by systematically repositioning the tool to account for surface or tree bark irregularities. The table in Fig. 5b presents the mean and standard deviation for the measured static coefficient of friction across all the studied poles.
---	---

pole	material	\varnothing (mm)	μ_s
I	copy paper	315	0.48 ± 0.02
II	copy paper	250*	0.48 ± 0.02
III	concrete	350	0.55 ± 0.05
IV	bamboo tree guard	260*	0.63 ± 0.04
V	rough concrete	350	0.64 ± 0.07
VI	rubber pad	315	0.65 ± 0.03
VII	rubber pad	250*	0.65 ± 0.03
VIII	paper towel	315	0.69 ± 0.01
IX	paper towel	250*	0.69 ± 0.01
X	Carpinus betulus	360	0.77 ± 0.05
XI	Carpinus betulus	320	0.83 ± 0.05
XII	Carpinus betulus	280	0.85 ± 0.05
XIII	Prunus avium	265	0.98 ± 0.11
XIV	Prunus mahaleb	350	1.02 ± 0.08
XV	Catalpa speciosa	270	1.04 ± 0.15

- Page 12, line 863-864: How to build the motion capture system outdoor?

We did not build a motion capture system outdoors, which is indeed impractical. We simply used a video-tracking software (Physlets Tracker) to effectively track and extract speed and angle information from multiple colored markers placed on the robot. The statement's language is improved to avoid such a confusion.	Dynamic perching experiments With the trigger strategy set to release upon primary impact, we used the PercHug prototype to conduct a minimum of five perching experiments on each tree for the two selected nose configurations, i.e., the standard upturned nose and the elastic extension with $D = 0.233 \text{ N}\cdot\text{m}^2$. This resulted in a total of 80 perching trials, approximately 40 tests for each nose type across the six trees. We used a camera to record the perching events in slow-motion at 240 Hz. The speed and pitch angle data were extracted from the recorded footage using the Physlets Tracker software, by tracking the colored markers placed on PercHug.
---	---

- The title of Figure S5 should be “b” rather than “c”.

The figure caption has been corrected.

- Supplementary Video should be a dynamic video that is not convenient to be expressed in the text, but the authors put some pictures in the text into the Supplementary Video, which is unreasonable. It is suggested that the authors could refer to the content of the supplementary video in some high-level papers.

In our initial submission, we aimed to present a comprehensive narrative of our work through a single video. However, we understand and appreciate the suggestion that high-level papers may not commonly follow this practice. To address this concern, we have revised our approach by creating multiple supplementary videos, each tailored to convey specific messages related	Supplementary Videos Video S1. Dynamic perching – different nose types. This video shows successful perching experiments with PercHug on various trees, using the standard upturned and extended elastic nose types. Video S2. Dynamic perching – wing release timing and hooks effects. This video shows the effects of unlatching strategy (primary vs. secondary impact release) and hooks effectiveness on perching success. Video S3. Dynamic perching – targeting and angular misalignments. This video highlights how mistargeting, lateral errors, and angular misalignments in the approach trajectory can result in perching failure.
--	---

to the results and discussion. None of these videos include pictures in the text. They are referenced in the revised manuscript, aligning each video with relevant discussions on aspects such as real-time and slow-motion experiments, nose variation, wing-release time and hooks effects, and failures due to poor approach conditions.

In all dynamic perching experiments, we hand-launched the robot toward the trees. A perching trial was considered successful if it involved four distinct phases of gliding, reorienting at impact, wrapping the wings, and staying perched on the tree (see Fig. 6a and Supplementary Video S1). The efficacy of hooks and the critical role of unlatching time in increasing perching success became evident during our preliminary investigations (Supplementary Video S2). Unlike in static perching, the robot often

and 180 ms, respectively. These results confirm the rapid dynamics of the perching maneuver and underscore the importance of timing the unlatching strategy to release at the primary impact.

Perching success is also significantly influenced by the quality of approach conditions. Impacting the target pole with the nose at its centerline emerges as a critical factor, as deviations in lateral direction or angular errors can lead to failure (Supplementary Video S3). Substantial lat-

Reviewer 2 Comments

- I believe in the entire manuscript, the authors have not adequately investigated the contribution of the hooks. Early on, the manuscript indicates the presence of hooks at the wing tips. However, in the subsequent modeling, the hooks were ignored (which is fine, but it should be made a little clearer). Then, in the dynamic perching validation, I believe the hooks were present. This distinction should be clearly mentioned. Then, the authors claim that the results match the models. Will they still match the models if the hooks were absent? It would be informative and helpful if the usefulness of the hooks is systematically verified.

We thank the reviewer for their insightful comment. Recognizing the oversight in our original draft, we have carefully revised the manuscript to explicitly state the presence or absence of hooks in both the modeling and experimental sections. They were indeed excluded from static experiments, which could introduce a very high randomness to the payload capacity based on manual placement and engagement with tree barks. This decision, along with the effects of hooks are now added to the revised manuscript. We have expanded the dynamic validation section to include a discussion on how hooks help increase success rate, as well as exemplifying their effectiveness through a new supplementary video (S2). This should now provide a better understanding of their contribution to dynamic perching. We believe that these additions clarify the distinction between modeling assumptions and experiments while addressing the concerns about the verification of the hooks' usefulness.

from Static Perching:

In order to understand the most suitable dimensions, we developed a wing-wrapping model (Fig. 4a; see Supplementary Fig. S3 and Methods for details), and validated it by conducting pole-hugging experiments with the predicted wing design. The model applies to static perching with hook-less wings, referring to the situation when the

from Experimental validation with PercHug:

hand-launched against trees. PercHug weighed 550 g (see Fig. 2c for the weight distribution), including the unlatching mechanism, bistable backup trigger, and folding wings with hooks (Fig. 2d-g), as well as a reinforced tail

launched the robot toward the trees. A perching trial was considered successful if it involved four distinct phases of gliding, reorienting at impact, wrapping the wings, and staying perched on the tree (see Fig. 6a and Supplementary Video S1). The efficacy of hooks and the critical role of unlatching time in increasing perching success became evident during our preliminary investigations (Supplementary Video S2). Unlike in static perching, the robot often encountered the issue of sliding down the pole with wings wrapped and a challenge to come to rest. The incorporation of hooks mitigated this problem significantly and was observed to be effective in over one-third of the successful trials. Hooks, through their randomized engagement with surface asperities, facilitated a rapid deceleration, effectively bringing the robot to a swift stop. We also in-

from Methods – Model assumptions:

The model does not account for the wing hooks, a simplification driven by the unpredictable nature of the interactions resulting from random engagement of hooks with pole surfaces or tree barks. Furthermore, it only supports symmetrical wing designs, but with a possibility of using uneven segmentation per wing. Even though the

from Supplementary Material:

Video S2. Dynamic perching – wing release timing and hooks effects. This video shows the effects of unlatching strategy (primary vs. secondary impact release) and hooks effectiveness on perching success.

- Related to the comment above, the fact that the static perching experiments were carried out with hook-less perching wings should be clear in the main text (not just in the Methods).

As mentioned, we have clarified in the main text, not just in the Methods section, that the static perching experiments were conducted without the use of hooks, providing a more transparent presentation of the experimental conditions.	Static perching The sizing and segmentation of the folding wings define the range of pole diameters on which the robot can perch. In order to understand the most suitable dimensions, we developed a wing-wrapping model (Fig. 4a; see Supplementary Fig. S3 and Methods for details), and validated it by conducting pole-hugging experiments with the predicted wing design. The model applies to static perching with hook-less wings, referring to the situation when the wings are already wrapped around the vertical pole.
--	--

- From my understanding, the force estimation (under Methods: Flight kinematics and impact force estimation) may not be technically correct. Let me elaborate and see if you disagree (I'm happy to change my view).
 - >> In short, what you claim to care about is the acceleration of the robot in the body frame. This is because you finally use a_i in (7), that is $F = m \cdot a_i$. Hence, why do you not just numerically take the derivatives of the position twice?
 - >> Instead, you first calculated the velocity of the robot, in the body-fixed frame, by reprojecting \dot{P} . This becomes V in (2).
 - >> The derivative of V in (6) is the acceleration in the body-fixed frame. This is not " a " in (6). In fact, $F = m(dV/dt + \Omega \text{ cross } V)$. This is force in the body frame. However, I believe you want to obtain force in the inertial frame.
 - >> Furthermore, this section is not extremely well organized. For example, you never use (3) (you only use the rotation matrix and euler angles, not the angular velocity). The sentence starting with "Any rigid body, such as flying robot, ..." at the end of the first paragraph and the sentence "Multiplying any vector defined in ..." are not really needed (they are very fundamental).
 - >> The recalculated accelerations may not be that different from the previous values, but it is important to make sure that the paper is technically sound.

We understand the concerns raised about the technical correctness of our force estimation method. Originally, we employed these equations to transform variables between raw tracking data in the inertial frame and body-fixed states, facilitating a comprehensive analysis of various variables for potential relationships to the tested parameters. Upon revisiting the original representation of the section, we concur with the reviewer's assessment that the complex body-fixed frame calculations and the use of certain equations, such as angular velocity, did not seem necessary. The	Here, we present the mathematical expressions used to estimate the robot's state variables and impact parameters. These estimations rely on the tracking data obtained from the motion capture system (OptiTrack) during the reorientation experiments. In our notation, subscripts B and I denote variables in the body-attached and inertial reference frames, respectively, and subscript i designates the magnitude of parameters at primary impact. We assume the robot is a rigid body, defined in space through three position and three linear velocity states for
--	--

Coriolis term was also correctly pointed out as missing in the acceleration equation. Consequently, we have thoroughly reorganized the section and equations to ensure a more technically sound and streamlined presentation. We have removed sentences on fundamental knowledge while providing clarifications on the notation, the use of state variables, and the estimation of the impact force/acceleration.

As a side note for reviewer's reference, we can confirm that the value of the force is almost unaffected if approximated by taking derivative of the position twice in inertial frame or using the body-fixed variables. This is because \mathbf{a}_B is inherently the acceleration in the inertial frame, projected at each instance of time onto the moving body-frame axes.

respectively. The derivative of the robot position $\vec{\mathbf{P}}_I$ ($\vec{\mathbf{V}}_I$) and attitude angles $\vec{\mathbf{T}}$ are numerically calculated from the tracked data using the central difference method.

For each reorientation experiment, we determine the primary impact phase (highlighted in red in Fig. 3b) based on the abrupt change in the state values of translational velocity u and angular velocity q (almost equivalent to the pitch rate $\dot{\theta}$, according to Eqs. (3) and (5), when the roll angle ϕ is nearly zero). For successful trials, the end of the reorientation maneuver, or approximately the onset of the secondary impact, is characterized by reaching the maximum pitch angle θ (almost 90°). We estimate the maximum primary impact force, denoted as F_i , by analyzing the change in acceleration profile during the primary impact. If m denotes the total mass of the vehicle and a_i is the magnitude of the maximum acceleration data point within the primary impact phase, then

$$F_i = ma_i . \quad (6)$$

The acceleration can be estimated by numerically differentiating the speed profile using the central difference scheme, either from the inertial velocity $\vec{\mathbf{a}}_I = \dot{\vec{\mathbf{V}}}_I$ or the body-fixed variables

$$\begin{aligned} \vec{\mathbf{a}}_B &= \dot{\vec{\mathbf{V}}}_B + \vec{\mathbf{\Omega}}_B \times \vec{\mathbf{V}}_B , \\ \vec{\mathbf{a}}_B^n &= \frac{\vec{\mathbf{V}}_B^{n+1} - \vec{\mathbf{V}}_B^{n-1}}{2\Delta t} + \vec{\mathbf{\Omega}}_B^n \times \vec{\mathbf{V}}_B^n , \end{aligned} \quad (7)$$

where $n + 1$ and $n - 1$ represent neighboring data points to point n , and Δt is the time step (equivalent to $\frac{1}{240\text{Hz}}$). Note that for the correct approximation of the impact force, the effect of gravity has to be removed, such that $a_i = |\vec{\mathbf{a}}_I - \vec{\mathbf{g}}|$ or $a_i = |\vec{\mathbf{a}}_B - \mathbf{R}_{BI}\vec{\mathbf{g}}|$, where the acceleration due to gravity is $\vec{\mathbf{g}} = [0 \ 0 \ 9.81]^\top \text{m/s}^2$.

Less Critical Comments

- Page 2, Line 120: "This gripping force, combined with the friction along the vertical axis resulting from the gravitational force, ..." >> I'm confused. What do you mean when you state that the gravitational force is resulted from the friction?

We thank the reviewer for pointing this out. The statement was confusing and incorrect, and we have removed "resulting from the gravitational force". It is indeed the magnitude of the gripping force that affects friction, not the gravitational force.

- Page 5, Line 242: I totally understand why the authors state that the maximum angular limit is 180 deg (which may sounds intuitive), but is this really a theoretical limit? I imagine is the friction coefficient is very high, perching should be feasible at larger angles (at least in theory)?

- Fig. 5c, please define "sensitivity".

- Fig. 6d, it would be useful to include absolute numbers with the success rates.

We have moved the table in Fig. 6d to the supplementary material, following the suggestion of Reviewer 1, and have included absolute numbers along with timing metrics for completeness. Additionally, we have rectified a minor numerical error in the total success rates, which is reflected throughout the revised manuscript.		Table S1 Dynamic perching statistics with PerchHug. The table shows perching outcomes on different trees, characterized by their diameters \varnothing and static friction coefficients μ_s. The presented data includes perching success rates, expressed as percentages and actual number of experiments conducted, with the standard upright and the extended elastic nose configurations. Additionally, the timing metrics for triggering and wrapping events are provided, measured from the impact instance.										
		tree	\varnothing (mm)	μ_s	success rate			triggering (ms)		wrapping (ms)		
			standard	elastic		standard	elastic	standard	elastic			
X	360	0.77	50%	(3/6)	38%	(5/13)	26 ± 5	39 ± 19	175 ± 33	163 ± 28		
XI	320	0.83	75%	(3/4)	25%	(1/4)	30 ± 6	42 ± 14	146 ± 18	150 ± 0		
XII	280	0.85	100%	(4/4)	86%	(6/7)	24 ± 5	31 ± 12	119 ± 16	138 ± 13		
XIII	265	0.98	100%	(4/4)	25%	(1/4)	23 ± 7	42 ± 10	164 ± 21	200 ± 0		
XIV	350	1.02	67%	(2/3)	67%	(2/3)	26 ± 6	24 ± 6	150 ± 6	131 ± 9		
XV	270	1.04	60%	(3/5)	14%	(1/7)	27 ± 10	40 ± 15	192 ± 19	171 ± 0		
					73%	(19/26)	42%	(16/38)	26 ± 7	37 ± 15	156 ± 30	152 ± 25

- Page 7, Line 335, the number 20 ms delay looks more like 25 ms in Fig. 6b.
- Page 9, Line 542, Dyneema is a tradename. It might be useful to replace that with "composite fiber? (Dyneema)".
- Page 9, Line 520, "tough" >> "tough".

The above corrections have been made.

- Discussion, I suppose the ability to take off afterwards should be in the future direction?

We appreciate the reviewer for pointing out this aspect, which was indeed missing from our discussion. In response, we have addressed this gap in the revised manuscript by adding details on the limitations of the platform and potential solutions, along with their associated challenges for integrating new technologies, including unperching. Please refer to the main text for the full discussion.	While we successfully demonstrated a fully passive pole perching method, our platform currently lacks flight and targeting capabilities prior to perching, along with the ability to unperch and take flight again. To address ver. Given the preloaded and hugging design of the wings, a crucial enhancement to the system involves transforming the latch-release mechanism into a revertible powered subsystem, enabling active control over wing opening and closing. This modification unlocks a range of new functionalities. First, it can facilitate thrust-assisted climbing by slightly opening the wings to loosen the gripping force, allowing movement across the pole. A quick release capability enables a seamless return to perching mode upon reaching the target spot on the pole. Second, it facilitates full wing opening for unperching, and third, aiding recovery from a failed perching event identified by the IMU to prevent a fall to the ground. Executing a controlled recovery maneuver involves reopening the wings quickly, coupled with using thrust and adequate control authority from aerodynamic surfaces to regain stable flight. Overcoming the high spring forces at the wing joints presents a challenge in designing this active wing opening mechanism, which necessitates using a slow high-torque motor to pull onto the tensioning wires given their short moment arm with the springs. Extending this arm and guiding the wire across higher points on the wings can alleviate this issue. Additionally, the slower wing opening rate compared to wing closing requires a solution for remaining attached to the pole during unperching. Potential strategies involve utilizing thrust for momentary hovering, incorporating retractable hooks under the fuselage for engagement, or utilizing materials with bidirectional friction properties to facilitate upward movement while preventing downward falls. These advancements will mark a pivotal step in over-
--	---

Reviewer 3 Comments

- The authors present a perching solution for small fliers in which a fixed-wing flier performs a controlled pitch-up maneuver and wraps its wings around a slim tree trunk or approximately vertically aligned branch in a hugging grasp. The maneuver just prior to landing is not dissimilar to that from Lussier-Desbiens et al. [2012] although the enveloping grasp is certainly more secure than hanging from feet with spines as in the previous work.

In summary, the paper presents an interesting alternative concept for perching on approximately vertically oriented tree trunks and branches. The references to previous related work and the dynamic modeling are appropriate and provide a good argument for future researchers to consider the authors' proposed approach. The overall writing style is clear and the illustrations are excellent, giving the reader a clear sense of what the wing-wrapping glider can do. The video and supplementary material are welcome.

Nonetheless, the work is in an early stage of development with respect to a complete solution. To put the work in perspective, the authors should talk about some of the extensions and how they would be accomplished. For comparison, if this work were submitted to IEEE, it would likely be considered suitable for ICRA or IROS, and potentially for RA-Letters with some attention to the items below, but not for TRO or a similar journal.

We appreciate the reviewer's meticulous evaluation of our manuscript. While acknowledging that our work is in the developmental stage, it introduces a groundbreaking crash-perching method at high speeds, featuring impact force redirection for functional use—a notable departure from the common, yet potentially dangerous, pitch-up maneuvers for speed reduction observed in the referenced work. It is crucial to emphasize that the proposed method signifies a significant leap forward in UAV capabilities. We firmly believe its innovative nature and potential applications warrant the attention of a broader audience through journal dissemination. Additionally, the paper constitutes a contribution because it provides the foundational knowledge required to develop future control schemes. We have taken the reviewers' comments into account and implemented substantial enhancements in the revised manuscript, including the introduction of a simple dynamic model as well as additional considerations for future developments and the integration of new functionalities into the existing platform. These modifications not only address the concerns but also markedly reinforce our claims and findings, making a compelling case for the uniqueness and importance of our work.

Operational Questions and Concerns

- The plane in this paper is a glider. Is this proposed as the long term solution? Will gliders be released from a powered plane and then controlled to find perching sites? Alternatively, if the long-term solution is that wing wrapping will be used with a powered plane, where will the propeller go? Perhaps a rear propulsion scheme or with wing-mounted propellers? If the authors do not intend for the gliders to be used as the long term solution they should comment on this.

We are thankful to the reviewer for highlighting these important considerations. We deliberately focused on a simple gliding platform here to facilitate effective system characterization in our tests without introducing unnecessary complexities with a motor. However, this platform serves as an interim solution. Our vision involves transitioning to a powered system for enhanced capabilities. Future developments will integrate avionics to enable powered flight and address current limitations. The motor placement would certainly have to be in the back, such as the commonly used rear fuselage-mounted propulsion scheme. These clarifications and considerations are now reflected in the revised manuscript's discussion section for a more comprehensive overview.	While we successfully demonstrated a fully passive pole perching method, our platform currently lacks flight and targeting capabilities prior to perching, along with the ability to unperch and take flight again. To address these shortcomings, future developments based on this work will integrate avionics and proper control surfaces to enable autonomous flight and perching. Ongoing efforts (IMU). Additionally, the propulsion scheme is a critical consideration, favoring a rear fuselage-mounted one due to impracticality in mounting propellers on the nose or wings. The incorporation of a propulsion system also holds potential for further enhancing perching success by utilizing the thrust to push toward the pole and against gravity to reduce the bounce-off effect during the landing maneuver. Given the preloaded and hugging design of the wings,
--	---

- The paper claims a 71% success rate for perches; this is actually not particularly high. What happens to the approximately 30% that fail to perch? Do they fall to the ground? If the plane were powered there could be an emergency recovery solution (as in some previous work on perching fliers) – but that seems less likely with a glider.

We appreciate the reviewer's attention to the success rate, and we respectfully differ on the interpretation of a 71% perching success rate as low, particularly given the entirely passive nature of the process. Achieving this success passively is a noteworthy accomplishment with such a sophisticated fast maneuver. Regarding the roughly 30% failure rate, we recognize the raised concern. In response, we have commented on potential improvements that a powered system, or a full feedback control system, can bring to the success rate in the revised manuscript, such as utilizing the thrust force to assist in the perching maneuver or a potential recovery method when coupled with an active wing opening mechanism.	wings. The incorporation of a propulsion system also holds potential for further enhancing perching success by utilizing the thrust to push toward the pole and against gravity to reduce the bounce-off effect during the landing maneuver. Given the preloaded and hugging design of the wings, a crucial enhancement to the system involves transforming the latch-release mechanism into a reversible powered subsystem, enabling active control over wing opening and closing. This modification unlocks a range of new functionalities. First, it can facilitate thrust-assisted climbing by slightly opening the wings to loosen the gripping force, allowing movement across the pole. A quick release capability enables a seamless return to perching mode upon reaching the target spot on the pole. Second, it facilitates full wing opening for unperching, and third, aiding recovery from a failed perching event identified by the IMU to prevent a fall to the ground. Executing a controlled recovery maneuver involves reopening the wings quickly, coupled with using thrust and adequate control authority from aerodynamic surfaces to regain stable flight. Over-
---	---

- The paper covers relationships among nose geometry and stiffness and perching maximum forces and success rate for a variety of approach velocities, pitch angles, tree trunk diameters and materials. There is no discussion about the effects of errors in targeting in the lateral direction (i.e., making contact somewhat to the left or right of the centerline of the tree trunk) and no discussion about angular errors in the yaw and roll degrees of freedom. Controlled experiments for this prototype were evidently conducted indoors with motion capture, where such errors may be kept small. But the intended use is outdoors, where air currents will make such errors unavoidable. One such alignment error appears to occur at about 2:09 in the attached video.

Thus in the longer term it will be necessary to establish the “envelope” of initial conditions in targeting, as well as velocities, for all degrees of freedom.

We acknowledge the importance of targeting and angular misalignments, particularly in real-world scenarios. Systematically testing these cases is inherently challenging, given the hand-launching method of outdoor tests. However, we fully recognize the importance of establishing the envelope of initial conditions, and thus, we have revised the text to comment on these matters explicitly. Additionally, we have included and exemplified more of such cases in a new supplementary video (S3), providing further insights into potential issues under less-than-ideal conditions.	from Supplementary Material: Video S3. Dynamic perching – targeting and angular misalignments. This video highlights how mistargeting, lateral errors, and angular misalignments in the approach trajectory can result in perching failure. from Experimental validation with PercHug: dynamics of the perching maneuver and underscore the importance of timing the unlatching strategy to release at the primary impact. Perching success is also significantly influenced by the quality of approach conditions. Impacting the target pole with the nose at its centerline emerges as a critical factor, as deviations in lateral direction or angular errors can lead to failure (Supplementary Video S3). Substantial lateral offset risks impacting the tree with the wings rather than the nose, resulting in no reorientation, and hence, failure. Moreover, even if the impact is on the nose but off-centered, there is a potential scenario where one wing wraps around the trunk while the other remains too distant, leading to unsuccessful perching. Angular errors in all degrees of freedom can also contribute to failure. As expected, a low impact angle, which generally resembles a low pitch angle, leads to insufficient reorientation and a rebound off the tree. Furthermore, errors in roll and yaw can lead to a missing tail impact and subsequent failure. The rigid tail of PercHug plays a crucial role in halting the reorientation maneuver upon contact with the trunk when reaching a vertical position. This is evident from the pitch data, where the pitch increase ceases and levels off at around 90° due to the tail’s contact (Fig. 6b). In trials where the robot did not make contact with its tail, it continued to reorient beyond 90°, resulting in unsuccessful landing maneuvers. This finding aligns with the research
---	---

Detailed Comments on the Manuscript

- I find figure 3c confusing. Looking at the data and variations in the data I cannot discern a clear correlation among design parameters, success rate and maximum impact forces apart from the fact that success rate increases and variability decreases with beta. The claimed optimality of $D=0.233Nm^2$ is not strongly supported. The authors write: “The improved success rate may be a result of a longer moment arm from the impact point to the COG of the robot”, which is speculative. But this should be fairly straightforward to test either based on a kinematic simulation or empirically.

To address the concern, we have updated the colormaps in Fig 3c to enhance the visual representation. Besides the correlation that the reviewer has correctly noted, there is a direct linear correlation between impact force and speed due to the increased momentum. These are now clarified in our updated manuscript. Additionally, as per the reviewer's suggestion, we have included new equations to better highlight the role of the dominant parameters (nose offset) in reorientation dynamics. It is detailed in the revised Methods section and supports our claim about the impact of the moment arm. By referring to this equation in the main text, we clarify how an increased moment arm contributes to better reorientation performance, i.e., improved success rates. We invite the reviewer to refer to the revised text and figures for a comprehensive overview of the details.

Figure 3 Reorientation maneuver and performance with four different noses. a illustration of the concepts of unsuccessful reorientation, where

from **Inertial Reorientation**:

flexibility. Although the elastic nose with $D = 0.233 \text{ N}\cdot\text{m}^2$ improves the success rate at lower impact angles compared to the standard upturned nose, Fig. 3c also shows that they share similar amounts of impact force over the range of tested impact speeds. The observed improvement is associated with the elongated moment arm between the impact point and COG (Fig. 2a and Supplementary Fig. S2). According to the simplified reorientation dynamics (Eq. (9)), pitching acceleration is influenced by the rigid nose tip offset parameters. This effect is notable in the stiffest elastic nose extension that acts more like a rigid nose and transmits a greater force and moment to the COG, which causes a faster pitching acceleration (Fig. 3c). Moreover, Eq. (9) suggests that at lower pitch angles the vertical nose offset contributes more to the pitching moment, while the longitudinal offset has minimal effect. Hence, enhancing success at lower impact angles can be accomplished by increasing the vertical nose tip offset. The data presented aids in estimating impact forces for similar-sized robots at varying weights, offering valuable insights for airframe mechanical design and structural analysis.

from **Methods**:

To better comprehend the dynamics of reorientation and identify the dominant parameters, we examine the angular acceleration induced by the impact force's moment. The rotational motion of the robot can be represented as

$$\vec{M}_B = \mathbf{I}_G \dot{\vec{\Omega}}_B + \vec{\Omega}_B \times (\mathbf{I}_G \vec{\Omega}_B), \quad (8)$$

where \mathbf{I}_G is the inertia tensor calculated about the body-fixed axes at the COG, and $\vec{M}_B = \vec{r} \times \vec{F}$ is the net moment about the body-fixed axes, resulting from the external force \vec{F} applied at a moment arm \vec{r} from the COG. If we express the nose tip offset relative to the COG as $\vec{r} = [l_{x_{off}} \ 0 \ -l_{z_{off}}]^T$ (see Supplementary Fig. S2) and assume the robot is flying toward a vertical target with only a pitch angle of θ (zero roll and yaw angles, and no angular speeds), then the moment equation due to the impact force F_i simplifies into the following pitching acceleration dynamics

$$\dot{q} = \ddot{\theta} = \frac{F_i (l_{z_{off}} \cos \theta + l_{x_{off}} \sin \theta)}{I_y}, \quad (9)$$

where I_y is the robot's mass moment of inertia about the lateral axis.

- A further discussion about the considerations for autonomous flight would make the paper stand out more. Beyond discussing that this is future work, some consideration for what sensors/planning/ etc method would enable autonomous perching would be useful to the reader.

We agree that this information is valuable for the reader. In our revised manuscript, we have elaborated on avionics integration for autonomous flight and the perching mission's planning phase. This encompasses vision-based pole detection, targeting, and autonomous navigation with GNSS and vision control methods. We have also cited relevant works to provide a more comprehensive understanding of how these goals can be achieved.	the ability to unperch and take flight again. To address these shortcomings, future developments based on this work will integrate avionics and proper control surfaces to enable autonomous flight and perching. Ongoing efforts involve sensor-based pole detection for perching, a powered grip loosening control for thrust-assisted climbing, and unperching, collectively contributing to a complete mission cycle. In the planning phase of a perching mission, a GNSS-receiver can provide means to autonomously navigate from far away to a target region within a few meters of accuracy [37]. Subsequently, a target pole can be detected using vision-based methods from tens or hundreds of meters away [38–40]. Switching to an accurate vision-based flight control method can then facilitate maneuvering to the desired spot, as demonstrated in [41]. The success or failure of the perching maneuver can be determined using data from an inertial measurement unit
--	---

- The wing cannot be automatically removed from the tree branch. What are some of the challenges in doing so? This also ties in with the need for a recovery strategy including how to detect failure and how to recover from it without falling to the ground.

Addressing wing removal challenges involves the need for quick and lightweight wing opening. We have proposed a high-torque wire-spooling motor as one option, but it can be slow and heavy. Implementing solutions like increasing the moment arm can help reduce this issue. However, instant wing opening remains a challenge. Ensuring perch stability during wing opening also poses difficulties. Our proposed strategies involve using thrust, retractable hooks, or bidirectional friction materials to tackle these challenges. To detect failure and trigger recovery without falling, we can monitor IMU data post-impact to distinguish between successful perching (remaining static) and unsuccessful ones (falling down).	The success or failure of the perching maneuver can be determined using data from an inertial measurement unit (IMU). Additionally, the propulsion scheme is a critical from aerodynamic surfaces to regain stable flight. Overcoming the high spring forces at the wing joints presents a challenge in designing this active wing opening mechanism, which necessitates using a high-torque motor, which can be slow and heavy, to pull onto the tensioning wires given their short moment arm with the springs. Extending this arm and guiding the wire across higher points on the wings can alleviate this issue. Additionally, the slower wing opening rate compared to wing closing requires a solution for remaining attached to the pole during unperching. Potential strategies involve utilizing thrust for momentary hovering, incorporating retractable hooks under the fuselage for engagement, or utilizing materials with bidirectional friction properties to facilitate upward movement while preventing downward falls. These ad-
--	--

- A low impact angle leads to lower success rate. How could the design be improved to improve perching at low impact angle?

The revised manuscript now includes a simplified equation of the reorientation dynamics, which provides a clearer understanding of the correlation between the center of gravity (COG) location, relative nose tip placement, and impact angle. This model effectively proves the potential improvements in success rate at lower impact angles through an optimized nose tip placement. Additionally, our new discussion on a futuristic powered system outlines how success rates can be further improved by employing a feedback-controlled thrust force or an improved approach trajectories.

from **Supplementary Material:**

from **Methods:**

no angular speeds), then the moment equation due to the impact force F_i simplifies into the following pitching acceleration dynamics

$$\dot{q} = \ddot{\theta} = \frac{F_i (l_{z_{off}} \cos \theta + l_{x_{off}} \sin \theta)}{I_y}, \quad (9)$$

from **Inertial Reorientation:**

Moreover, Eq. (9) suggests that at lower pitch angles the vertical nose offset contributes more to the pitching moment, while the longitudinal offset has minimal effect. Hence, enhancing success at lower impact angles can be accomplished by increasing the vertical nose tip offset. The data presented aids in estimating impact forces for similar-sized robots at varying weights, offering valuable insights for airframe mechanical design and structural analysis.

from **Discussion:**

wings. The incorporation of a propulsion system also holds potential for further enhancing perching success by utilizing the thrust to push toward the pole and against gravity to reduce the bounce-off effect during the landing maneuver. Given the preloaded and hugging design of the wings, a crucial enhancement to the system involves transforming the latch-release mechanism into a reversible powered subsystem, enabling active control over wing opening and closing. This modification unlocks a range of new functionalities. First, it can facilitate thrust-assisted climbing by slightly opening the wings to loosen the gripping force, allowing movement across the pole. A quick release capability enables a seamless return to perching mode upon reaching the target spot on the pole. Second, it facilitates full wing opening for unperching, and third, aiding recovery from a failed perching event identified by the IMU to prevent a fall to the ground. Executing a controlled recovery maneuver involves reopening the wings quickly, coupled with using thrust and adequate control authority from aerodynamic surfaces to regain stable flight. Over-

Minor Points

- Fig. 5c refers to error bars as ‘sensitivity’ - a better word would be variability (perhaps 1 standard deviation?).

We have removed the error bars to avoid any confusion, as they did not correspond to the standard deviation. In our revision, we have added more details to the caption and under the Methods section to clarify this point. Specifically, each static test was repeated five times with the payload increased by 100g calibration weight increments at the COG. In all the tests, the robot consistently held a similar payload on every pole and dropped with the addition of the next calibration weight, indicating that the actual maximum payload value is within a 100g range of the reported values.	Static perching experiments We validated the predicted wing design (see Wing segmentation sizing for configuration selection) with experiments of a physical prototype weighing 325 g attached to poles of different sizes and materials. The testing strategy involved placing the prototype robot, equipped with hook-less perching wings, on various poles and trees within the range specified by the model. We then gradually added calibration weights of 100 g each at the COG until the point of sliding down. We recorded the maximum weight that the prototype could sustain before falling as the static payload capacity (Fig. 5c). The measurements were repeated five times for each pole and showed no variation. The surface
Figure 5 Model validation with static perching experiments. a Close-up pictures of the surfaces of the poles used in the static perching experiments. b List of poles and their specifications in order of increasing friction coefficient. The "*" symbol denotes poles with a diameter smaller than the model's predicted minimum value of 265 mm. These cases were analyzed since the model was found to be valid even outside the previously mentioned diameter range, provided that the considered diameter is close to the limit. c Comparison of model predictions and actual measurements for the maximum static payload capacity. The estimated payload started with measurements at the weight of the prototype alone (325 g) and incremented by 100 g steps until failure (see Methods for more details). The insignificant discrepancies between experiment and	

- Figure 4 and related discussion: What exactly are the min and maximum wrap angles? It looks like the limits are from 180 degrees up to some number less than 360 degrees? What determines the maximum wrap angle? Further, in Fig. 5c - why does the solution sometimes work for diameters that are smaller than the theoretical smallest diameter (250 mm vs. 265mm)?

The maximum wrap angle occurs when the robot's wings are fully wrapped around the pole, and the wing tips meet at a point on the opposite end. This angle is slightly less than 360deg because of the way we define it (based on the contact locations along the outermost segments and the pole). The specific value depends on the number and lengths of the segments. We have now provided it for our selected segmentation size and the concepts are better clarified in the main text and under Methods. Regarding the smallest diameter, it is not a "theoretical" limit but rather a "practical" one, reflective of our model assumptions. We have replaced these terms to accurately	b  Figure 4 Static wing-wrapping model and pole gripping performance. a Free on a pole. b The practical limits of the pole diameters the robot can perch on. Similarly, here we assume that the maximum pole diameter on which the UAV can practically perch corresponds to a wing-wrapping angle (θ_w) of 180°, and the minimum pole diameter corresponds to a size that prevents the two wingtips from overlapping, (Fig. 4b). This wing-wrapping angle is defined by the portion of the pole covered by the folded wings, between the two contact locations along the outermost segments. Its maximum value depends on the wingspan and the number and lengths of the segments, approximately reaching 290° for our selected segmentation size. For the wings to remain wrapped, the normal
--	--

convey this distinction. On smaller diameters than 265mm, the model assumptions are no longer valid as either one wing has to overlap the other (unrealistic in dynamic perching) or the robot needs to slightly tilt in the vertical direction while the wingtips are touching, effectively reducing the wrapped diameter. The model predictions become further invalid as we further deviate from the minimum diameter. These corner cases fall outside the design scope, but are now better explained in the revised manuscript.

ods used to estimate coefficient of static friction). The tested objects included two indoor poles with diameters 250 mm and 315 mm, covered in copy paper, rubber pad, and paper towel (designated as poles I, II, and VI-IX), a 260 mm bamboo tree guard (IV), smooth and rough concrete columns of 350 mm (III and V), and six trees (X-XV) with diameters ranging from 265 mm to 360 mm. **Note that diameters smaller than 265 mm invalidate the model assumptions, as either one wing has to overlap the other or the robot needs to slightly tilt in the vertical direction while the wingtips are touching, effectively reducing the wrapped diameter. While such corner cases fall outside the design and modeling scope, we still considered a few pole sizes slightly below this threshold as results matched well. The model predictions become further invalid as we deviate further from the minimum diameter limit.**

- The static modeling seems mainly to inform us that the drone will not fall once it has successfully wrapped around a tree. It seems that the majority of failures will arise from dynamic interaction between the drone and the tree and an analysis of this dynamic interaction would be more informative. It may lead to insights regarding trajectories, spring stiffnesses for the torsional springs and triggering sensitivity that will increase the probability of success despite variations in targeting and velocity, as will occur in the field.

We agree with the reviewer that dynamic aspects of the interaction between the drone and the tree is equally important, if not more than the static model. The new simplified dynamic model we have introduced gives insight into effective design enhancements for success rate improvement, but does not consider such aspects. Particularly for trajectory planning, triggering sensitivity, and wrapping dynamics, a more sophisticated model is imperative. Such a model would need an accurate representation of aerodynamic forces, stall effects, actuator dynamics, nonlinear contact mechanics of impact and friction, and a better model of torsion spring behavior. In spite of these complexities, it is crucial to acknowledge that bridging the sim-to-real gap has been a significant challenge, especially for aerial systems with intricate and rapid dynamics, a concern that has garnered attention in recent studies. Developing and validating such a comprehensive model requires a dedicated study, which we are delving into in an upcoming work. We appreciate the reviewer's comment, and the revised manuscript and discussions now better emphasize the potential insights that can be gained from an improved dynamic model.

REVIEWERS' COMMENTS:

Reviewer #1 (Remarks to the Author):

The revised article has basically modified according to the previous comments, but there is also a problem that the overall success rate crash-perching on tree trunks has changed from 71% to 73%, and is not reflected the reason in the article (Supplementary Table S1 for a detailed breakdown).

Reviewer #2 (Remarks to the Author):

I appreciate the thorough revisions made in response to my feedback. The manuscript is significantly improved through the additions and clarifications on the hooks' contribution, friction modeling, and force calculations. The supplementary videos also provide helpful visual demonstrations. Overall, the technical soundness is enhanced and my concerns have been adequately addressed.

Reviewer #3 (Remarks to the Author):

The authors have made a solid effort to address the comments raised by the reviewers and the clarity of the modified sections of the article is improved. I do not have further detailed comments at this time.

That said, I think the authors acknowledge that this is (useful) preliminary step toward a robotic system for autonomous or semi-autonomous fixed-wing perching on tree trunks and poles.

Crash-Perching on Vertical Poles with a Hugging-Wing Robot

Mohammad Askari, Michele Benciolini, Hoang-Vu Phan, William Stewart,

Auke J. Ijspeert and Dario Floreano

The authors thank all the reviewers for their approval of the revisions made to the manuscript. Their comments were invaluable and helped make the manuscript contributions more solid. Here, the response to Reviewer 1's recent concern is given.

Reviewer 1 Comment

- The revised article has basically modified according to the previous comments, but there is also a problem that the overall success rate crash-perching on tree trunks has changed from 71% to 73%, and is not reflected the reason in the article (Supplementary Table S1 for a detailed breakdown).

We sincerely appreciate the reviewer's diligent examination of the revised manuscript. The reviewer has correctly pointed out to the change in the reported perching success rate. During the previous revision phase, we identified a minor numerical error in the calculation of the total success rates, which we rectified and reflected throughout the manuscript. We had acknowledged this correction in our responses to Reviewers 2 and 3 comments in the previous response letter. However, we missed to bring this matter also to Reviewer 1's attention, and we apologize for any confusion this may have caused.